# Global burden, projections, and causal factors of maternal sepsis and other maternal infections: A comprehensive epidemiological and mendelian randomization study

Anqi Jiang[1‡], Siying Duan[2‡], Shuyun Wu[1], Wenkui Yu[1*], Xiaohui Liang🄳[1*]

**1** Department of Critical Care Medicine, Nanjing Drum Tower Hospital, Affiliated Hospital of Medical School, Nanjing University, Nanjing, Jiangsu, China, **2** Emergency Department, The Eighth Affiliated Hospital, Sun Yat-sen University, Shenzhen, Guangdong, China

‡ These authors share first authors.
* yudrnj@163.com (WY); 439397013@qq.com (XL)

## Abstract

### Background

Maternal sepsis and other maternal infections (MSMI) remain major contributors to global maternal morbidity and mortality. However, the integration of epidemiological trends with causal inference evidence remains limited.

### Methods

Using data from the Global Burden of Disease (GBD) 2021 study, we assessed temporal trends in MSMI burden from 1990 to 2021 and projected future patterns using ARIMA and Bayesian age–period–cohort (BAPC) models. In parallel, we conducted a two-sample multivariable Mendelian randomization (MVMR) analysis to evaluate the causal effects of inflammatory biomarkers and related factors on MSMI risk.

### Results

Although age-standardized rates declined globally, absolute case numbers increased in low-SDI regions, largely driven by population growth. Forecasting results differed between ARIMA and BAPC models, reflecting distinct underlying assumptions regarding temporal dynamics. MVMR analysis identified inflammatory biomarkers, including CRP, IL-13, IL-10, RANTES, and NT-proBNP, as key causal factors associated with MSMI.

### Conclusions

This study provides the first integrated framework combining global disease burden analysis with multivariable MR. By linking population-level trends with causal

**Data availability statement:** All data analyzed during this study are included in this published article and its supplementary information files. The publicly available datasets analyzed during the current study are available in Tables and Supplementary Tables.

**Funding:** The author(s) received no specific funding for this work.

**Competing interests:** The authors have declared that no competing interests exist.

inference, our findings offer dual evidence to support targeted prevention strategies and advance precision public health interventions for MSMI.

## Author summary

This study found that while global rates have improved overall, the total number of cases has actually grown in poorer regions due to population increases. This tells us that progress is uneven and that future solutions must fit local realities. More importantly, we identified several natural immune-related markers in the blood-such as C-reactive protein and specific signaling molecules-that appear to directly raise the risk of maternal infections. These findings matter to scientists because they offer clear targets for future research and drug development. For non-scientists, especially public health officials and expectant families, our work shows that monitoring inflammation levels could one day help predict or even prevent these dangerous infections. By linking global patterns with genetic evidence, we provide a more complete roadmap for saving mothers' lives.

## Introduction

Maternal sepsis and other maternal infections (MSMI) are defined as sepsis occurring during the delivery or postpartum period, along with other infectious conditions closely associated with pregnancy. These encompass genitourinary tract infections (excluding sexually transmitted infections), obstetric surgical site infections, and breast infections related to childbirth and lactation. Such infections pose a serious threat to both maternal and neonatal health [1–2]. On the one hand, obstetric infections can lead to chronic pelvic inflammatory disease, ectopic pregnancy, infertility, and even maternal death. Among these outcomes, puerperal sepsis stands as one of the leading causes of maternal mortality. Global estimates indicate that approximately 10.7% of maternal deaths are attributable to puerperal sepsis [2]. On the other hand, these infections also contribute to short-term neonatal complications, such as intraventricular hemorrhage (IVH), respiratory distress syndrome (RDS), and necrotizing enterocolitis (NEC), and may result in long-term sequelae, including cerebral palsy and intellectual disability. Additionally, they increase the risks of preterm birth and fetal growth restriction. Epidemiological evidence suggests that nearly one million neonatal deaths annually are attributable to maternal infection or sepsis [3]. Although existing studies have delineated certain epidemiological characteristics of maternal sepsis and related infectious diseases, most findings are confined to specific geographic regions and frequently lack age-standardized methodologies, thereby limiting comparability across different countries and settings [4–7]. Furthermore, due to the complex interplay of age, period, and cohort effects, the independent contribution of each factor to morbidity and mortality risks remains poorly understood. Consequently, accurate prediction of MSMI incidence and mortality trends is of particular

importance, as it will furnish essential evidence for optimizing the allocation of medical resource, formulating targeted prevention strategies, and implementing effective clinical interventions.

The Global Burden of Diseases, Injuries, and Risk Factors Study 2021 (GBD 2021) provides a standardized and systematic assessment of 369 diseases and injuries and 87 risk factors across 204 countries and territories, thus offering essential data support for global public health research [8]. GBD 2021 data play a central role in numerous studies by facilitating the characterization of disease burden, analysis of temporal trends, future projections, and monitoring of health inequities, thereby forming a critical evidence base for public health policies and intervention strategies [9–10]. Using GBD 2021 data, this study calculated the age-standardized incidence and mortality rates of MSMI among women of reproductive age. Trends from 1990 to 2021 were analyzed, the independent effects of age, period, and birth cohort were evaluated, and the disease burden for the next 25 years was projected. These findings are intended to serve as a comprehensive reference for formulating targeted prevention and control strategies for MSMI at global, regional, and national levels.

Disparities in disease burden provide policymakers with an evidence-based foundation for implementing targeted interventions and preventive measures in specific populations. Nevertheless, patients affected by MSMI, along with their families, often seek to understand the underlying causes of these conditions to adopt effective prevention strategies. Therefore, it is essential to thoroughly investigate the risk factors contributing to MSMI for developing effective interventions aimed at reducing associated risks. Unlike traditional observational studies, which are susceptible to residual confounding and various biases, Mendelian randomization (MR) is an epidemiological approach that uses genetic variants-randomly allocated at conception-as instrumental variables to infer causal relationships between exposures and outcomes [11]. This method substantially reduces reverse causation and confounding biases often introduced by sociodemographic or behavioral factors in conventional observational studies. In this study, we applied a two-sample MR design to examine bidirectional causal relationships between various potential risk factors and MSMI. Furthermore, multivariable MR was employed to assess the independent effects of multiple significantly associated exposures on MSMI outcomes. The findings reveal causal links between several modifiable factors and MSMI, which may inform future preventive strategies and clinical guidance.

## Materials and methods

### Ethics statement

The FinnGen study protocol was approved by Coordinating Ethics Committee of the Hospital District of Helsinki and Uusimaa (number HUS/990/2017), with all participants providing written informed consent [14].

### Overview and global data sources

A schematic overview of the study design is shown in Fig 1. Data pertaining to MSMI were obtained from the Global Health Data Exchange (GHDx) data query tool (https://vizhub.healthdata.org/gbd-results/). Case definitions for MSMI were aligned with the diagnostic criteria set forth in the International Classification of Diseases, Ninth and Tenth Revisions (ICD-9 and ICD-10) [12]. As previously defined [12], MSMI comprises two distinct entities [12]. Maternal sepsis is characterized by abnormal core body temperature ( < 36°C or >38°C) along with signs of shock, such as systolic hypotension (<90 mmHg) and tachycardia (>120 beats per minute). Other maternal infections refer to any non-AIDS-defining and non-sexually transmitted infectious conditions not considered epidemiologically linked to pregnancy. The latter category encompasses disorders such as gestational urinary tract infections, mastitis, candidiasis, and bacterial vaginosis.

We sourced comprehensive global data on health losses attributable to MSMI between 1990 and 2021, spanning 204 countries and territories and 21 regions. Key metrics included incidence, prevalence, mortality, years lived with disability (YLDs), years of life lost (YLLs), and disability-adjusted life years (DALYs), along with their corresponding age-standardized rates (ASIR, ASPR, ASMR, AS-YLD, AS-YLL, and ASDR). DALYs were derived as the sum of YLDs

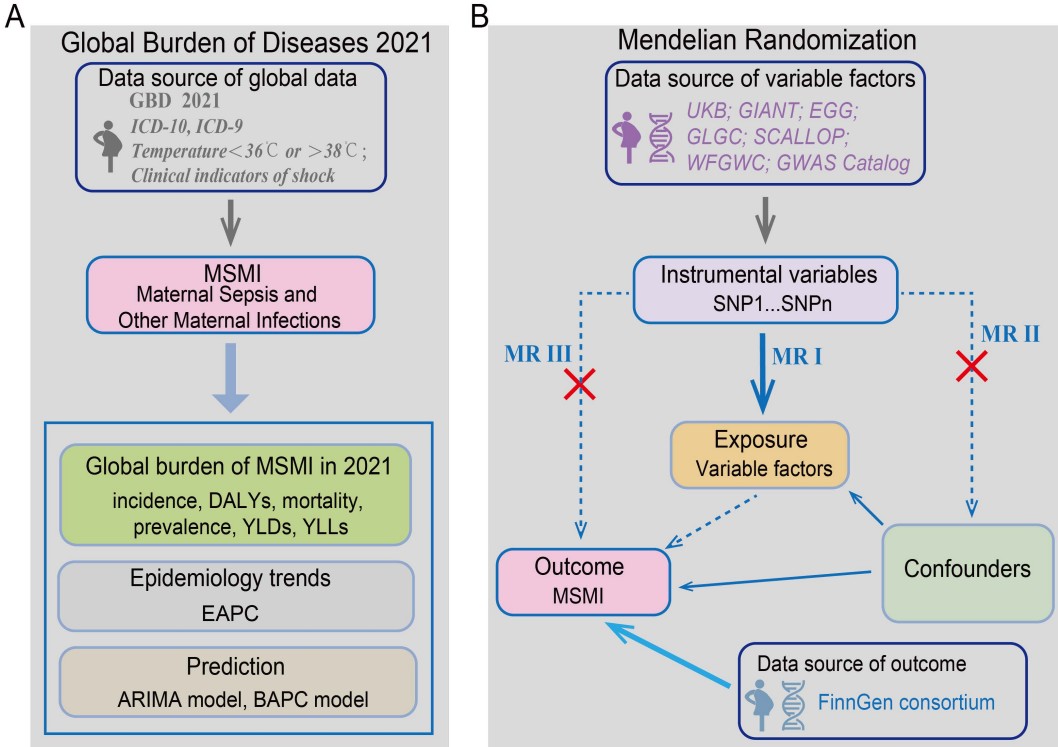

**Fig 1. Study Overview: Global Burden Analysis and Mendelian Randomization Design.** This flowchart outlines our two-pronged analytical approach. The Global Burden of Disease assessment was summarized in A, quantifying the worldwide impact of MSMI through metrics including incidence, DALYs, mortality, prevalence, YLDs, YLLs and epidemiology trends using EAPC. The MR framework was summarized in B, where genetic variants SNPs significantly associated with exposure phenotypes serve as instrumental variables **(IVs)**. The MR design rests on three core assumptions: **(I)** Relevance: SNPs must strongly associate with the exposure; **(II)** Independence: SNPs must be independent of exposure-outcome confounders; and **(III)** Exclusion restriction: SNPs must affect the outcome exclusively through the exposure, without alternative pathways. YLDs, Years Lived with Disability; DALYs, Disability-Adjusted Life Years; YLLs, Years of Life Lost. Methodological abbreviations: EAPC, Estimated Annual Percentage Change; ARIMA, autoregressive integrated moving average; BAPC, Bayesian age-period-cohort.

and YLLs. YLDs were calculated by multiplying the prevalence of each disease sequela by its disability weight, while YLLs were estimated based on cause-specific deaths multiplied by the standard life expectancy at the age of death [12]. Although global standardization efforts under the GBD framework have enhanced data comparability, disparities in socioeconomic status and variations in reproductive-age population structures may still influence data quality. To address potential confounding from these factors, stratified analyses of MSMI-related health loss were conducted by Socio-demographic Index (SDI) and by maternal age group. The SDI is a composite measure of sociodemographic development that integrates income, educational attainment, and fertility rates [12]. Based on 2021 SDI quintiles, all countries and territories were classified into five development strata: low, low-middle, middle, high-middle, and high, where elevated SDI reflects advanced socioeconomic conditions [12]. Additionally, temporal trends in MSMI burden were evaluated across detailed age categories as defined in GBD 2021: 10–14, 15–19, 20–24, 25–29, 30–34, 35–39, 40–44, 45–49, and 50–54 years [13]. Complete datasets supporting these analyses are available in S1-S6 Tables.

## MR study design

A two-sample MR analysis was employed to assess the potential causal effects of multiple exposures on MSMI. The analysis was grounded in three core MR assumptions: (i) the genetic instruments exhibit strong associations with the

exposures of interest; (ii) these instruments are independent of potential confounders; and (iii) they influence the risk of MSMI only via the exposure pathways (Fig 1). As all genetic and outcome data were derived from previously published studies with ethical approvals [13], no additional informed consent was required for the present study.

### MR date sources

Genome-wide association study (GWAS) summary statistics for MSMI were obtained from the FinnGen Consortium [14]. The analyzed variable factors encompassed five major categories:

- **Basic characteristics**: birth weight, body mass index (BMI);

- **Medical history:** previous smoking status, history of stillbirth, spontaneous miscarriage or termination, number of pregnancy terminations, medication use for cholesterol, blood pressure, diabetes, or exogenous hormones (e.g., cholesterol-lowering medication);

- **Routine blood and serum metabolic biomarkers:** high-density lipoprotein (HDL) cholesterol, low-density lipoprotein (LDL) cholesterol, triglycerides (TG), total cholesterol levels or total cholesterol (TC), C-reactive protein levels or C-reactive protein (CRP), vitamin D levels (VitD), N-terminal prohormone of brain natriuretic peptide levels (NT-proBNP);

- **Serum inflammatory cytokines:** RANTES levels (RANTES), interleukin-13 levels (IL-13), interleukin-10 levels (IL-10), interleukin-1 receptor antagonist levels (IL-1Ra), interleukin-2 receptor antagonist levels (IL-2Ra), interferon gamma levels (IFN-γ);

- **Serum-specific biomarkers:** Pregnancy-specific beta-1-glycoprotein 11 (PSG11), Pregnancy-specific beta-1-glycoprotein 9 (PSG9).

Instrumental variables for these factors were sourced from multiple large-scale genetic consortia and databases, including the UK Biobank (UKB) [15], the Early Growth Genetics (EGG) Consortium [16], the Genetic Investigation of Anthropometric Traits (GIANT) Consortium [17], the Global Lipids Genetics Consortium (GLGC) [18], the NHGRI-EBI Genome-Wide Association Studies Catalog (GWAS Catalog) [19], the Systematic and Combined Analysis of Olink Proteins (SCALLOP) [20], and the Within Family GWAS Consortium (WFGWC) [21]. Detailed data source descriptions are provided in Table 1.

### Instrumental variable selection

Based on GWAS summary statistics, single-nucleotide polymorphisms (SNPs) associated with each exposure of interest were selected as instrumental variables. Independent SNPs were identified through a clumping procedure under strict linkage disequilibrium (LD) criteria ($P < 5 \times 10^{-8}$, $r^2 < 0.001$, clumping window = 10,000 kb). For unavailable SNPs in the outcome dataset, proxy variants were identified from the 1000 Genomes European reference panel using a high LD threshold ($r^2 > 0.8$), excluding SNPs without suitable proxies. Additionally, palindromic SNPs with intermediate allele frequencies were removed to prevent strand ambiguity. To enhance analytical robustness, SNPs with a minor allele frequency (MAF) < 0.01 were further excluded, thereby minimizing potential bias arising from low-frequency variants that often yield unreliable estimates in GWAS. The proportion of variance explained ($R^2$) for each risk factor by individual SNPs was quantified [13], and the strength of instrumental variables was evaluated using the F-statistic [22]. The F-statistic was calculated as $F = R^2 \times (N - 2) / (1 - R^2)$, where $R^2$ represents the proportion of variance explained by the genetic instruments and N is the sample size.

### Statistical analysis

Spearman's rank correlation analysis was employed to investigate the association between the Socio-demographic Index (SDI) and the burden of MSMI. The SDI is a composite indicator of a region's development level, ranging from 0 (lowest)

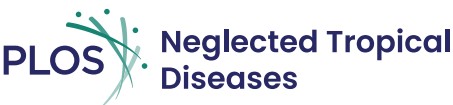

**Table 1. Data source of variable factors.**

| id.exposure | Exposure | Date source | Population |
|---|---|---|---|
| ieu-a-27 | Birth weight | EGG | European |
| ebi-a-GCST90095034 | Body mass index | GWAS Catalog | Hispanic or Latin American |
| ieu-a-94 | Body mass index | GIANT | European |
| ieu-b-4815 | Body mass index | Within family GWAS consortium | European |
| ukb-a-224 | Smoking status: Previous | UBK | European |
| ukb-b-12621 | Ever had stillbirth, spontaneous miscarriage or termination | UBK | European |
| ukb-a-355 | Number of pregnancy terminations | UBK | European |
| ukb-a-448 | Medication for cholesterol blood pressure diabetes or take exogenous hormones: Cholesterol lowering medication | UBK | European |
| ukb-b-17805 | Medication for cholesterol blood pressure diabetes or take exogenous hormones: Cholesterol lowering medication | UBK | European |
| ukb-b-18009 | Medication for cholesterol, blood pressure, diabetes, or take exogenous hormones: Blood pressure medication | UBK | European |
| ebi-a-GCST000755 | HDL cholesterol | GWAS Catalog | European |
| ebi-a-GCST000759 | LDL cholesterol | GWAS Catalog | European |
| ebi-a-GCST005068 | LDL cholesterol | GWAS Catalog | European |
| ebi-a-GCST90018741 | LDL cholesterol | GWAS Catalog | East Asian |
| ieu-a-781 | LDL cholesterol | GLGC | Mixed |
| ieu-b-4845 | LDL cholesterol | Within family GWAS consortium | European |
| ebi-a-GCST000758 | Triglycerides | GWAS Catalog | European |
| ieu-a-783 | Triglycerides | GLGC | Mixed |
| ieu-b-4849 | Triglycerides | Within family GWAS consortium | European |
| ieu-b-4850 | Triglycerides | Within family GWAS consortium | European |
| ebi-a-GCST90018754 | Total cholesterol levels | GWAS Catalog | East Asian |
| ebi-a-GCST90101747 | Total cholesterol levels | GWAS Catalog | Sub-Saharan African |
| ieu-a-782 | Total cholesterol | GLGC | Mixed |
| ebi-a-GCST005067 | C-reactive protein levels | GWAS Catalog | European |
| ebi-a-GCST90018730 | C-reactive protein | GWAS Catalog | East Asian |
| ebi-a-GCST005367 | Vitamin D levels | GWAS Catalog | European |
| ebi-a-GCST90012082 | N-terminal prohormone brain natriuretic peptide levels | GWAS Catalog | European |
| ebi-a-GCST004431 | RANTES levels | GWAS Catalog | European |
| ebi-a-GCST004443 | Interleukin-13 levels | GWAS Catalog | European |
| ebi-a-GCST004444 | Interleukin-10 levels | GWAS Catalog | European |
| ebi-a-GCST004447 | Interleukin-1-receptor antagonist levels | GWAS Catalog | European |
| ebi-a-GCST004454 | Interleukin-2-receptor antagonist levels | GWAS Catalog | European |
| ebi-a-GCST004456 | Interferon gamma levels | GWAS Catalog | European |
| prot-a-2400 | Pregnancy-specific beta-1-glycoprotein 11 | SCALLOP | European |
| prot-a-2408 | Pregnancy-specific beta-1-glycoprotein 9 | SCALLOP | European |

to 1 (highest), based on lag-distributed income per capita, average educational attainment, and total fertility rate. Given the non-normal distribution of burden indicators across countries, Spearman's rank correlation was selected as it is a non-parametric method that does not assume linearity or normality, and is robust to outliers. All correlation analyses were performed at the country level, including all 204 countries and territories in the GBD study for the year 1990 and 2021. For each pair of variables (SDI vs. each burden indicator), Spearman's correlation coefficient (r) and the corresponding two-tailed p-value were calculated. A p-value < 0.05 was considered statistically significant. All statistical analyses were conducted using R software (version 4.5.3) with the "cor.test" function [12]. Decomposition analysis was performed to quantify the contributions of demographic factors and epidemiological changes to the disparities in the burden of MSMI across different SDI quintiles in 2021. Using the Das Gupta decomposition method, the difference in outcome measure between each SDI quintile (low, low-middle, middle, high-middle, and high SDI) and the global average was partitioned into three components: (1) population structure (aging), reflecting differences in the age composition of populations; (2) population size (growth), reflecting differences in total population across SDI quintiles; and (3) epidemiological changes, reflecting differences in age-specific rates of the outcome measure. Absolute contributions (the number of cases attributable to each factor) and relative contributions (the percentage of the total disparity explained by each factor) were calculated for each SDI quintile. All decomposition analyses were conducted using R software (version 4.5.3) with the "**Dasgupta**" function from the "**decompose**" package [12].

The Estimated Annual Percentage Change (EAPC) serve as a key indicator for quantifying temporal trends in epidemiological metrics-such as incidence, mortality, or DALYs-over a specified period. It reflects the average annual rate of change, expressed as a percentage. In this study, the EAPC was applied to analyze trends in the ASR of MSMI from 1990 to 2021, based on data from the GBD 2021 database. EAPC was derived using a linear regression model applied to the natural logarithm of ASR values collected over consecutive years. The use of ASR mitigates confounding from temporal shifts in population age structure over time. The regression model is specified as: **ln (ASR) = α + β × year + ε**, where **ln (ASR)** is the natural log-transformed ASR, **year** represents the time variable, **α** is the intercept, **β** is the regression coefficient for time, and **ε** denotes the random error term. EAPC is calculated using the estimated slope **β** with the formula: **EAPC = (e^β - 1) × 100**. The 95% confidence interval (CI) is obtained from the standard error (SE) of **β**: **Upper limit = {exp (β + 1.96 × SE_β) - 1} × 100, Lower limit = {exp (β - 1.96 × SE_β) - 1} × 100.** A statistically significant increasing trend in ASR is concluded if both the EAPC and the lower limit of its 95% CI exceed zero, whereas a decreasing trend is inferred when the EAPC and the upper limit of the 95% CI are below zero.

To project future disease burden, two complementary modeling approaches were employed, including the autoregressive integrated moving average (ARIMA) model and the Bayesian age-period-cohort (BAPC) model. As a well-established method for epidemiological forecasting [23], the ARIMA model was fitted to historical time-series data from GBD 2021. After ensuring stationarity through differencing, the "auto.arima()" function was used to identify the optimal model configuration based on the lowest Akaike Information Criterion (AIC) and Bayesian Information Criterion (BIC). Model adequacy was confirmed via white noise testing (P > 0.05 for residuals). The "forecast" package was employed to predict future MSMI-related metrics at α = 0.05 significance. Simultaneously, the BAPC model captured complex temporal structures by integrating age, period, and cohort effects through Bayesian methodology, INLA algorithm for efficient prior and posterior distribution updating. A log-linear regression model was fitted for MSMI metrics as follows: **ln (ASR) = α + β_age × age + β_period × period + β_cohort × cohort + ε**, with Bayesian parameter estimated via the integrated nested Laplace approximation (INLA) algorithm for efficient prior and posterior distribution updating [24].

A two-sample MR framework was applied to investigate the causal relationships between variable factors and MSMI. After harmonizing effect alleles and directions using the "**harmonise_data**" function from the TwoSampleMR package in R software (version 4.1.0), MR analyses were conducted through the "**mr**" function, incorporating five complementary methods: MR-Egger regression (MR-Egger) [25], weighted median (WM) [25], inverse-variance weighted (IVW) [26], simple

mode, and weighted mode [26]. MR analyses were performed separately for each exposure factor, with the IVW method serving as the primary analytical approach for causal inference.

Heterogeneity among instrumental variables was assessed using Cochran's Q test via the "**mr_heterogeneity**" function. P-values > 0.05 indicated absence of significant heterogeneity, supporting the use of fixed-effect inverse-variance weighted (IVW) estimates. In this study, all Q test results were non-significant (P > 0.05), confirming homogeneity across variables [27]. Horizontal pleiotropy was evaluated using MR-Egger regression and MR-PRESSO through the "**mr_pleiotropy_test**" function. P > 0.05 in both tests indicated no evidence of directional pleiotropy, thereby ensuring causal estimate reliability [25,28]. Finally, causal directionality was verified using MR-Steiger analysis, which confirmed the presumed direction (exposure → outcome) by demonstrating greater variance explained in exposure than outcome. Sensitivity analysis was conducted using the "**mr_leaveoneout**" function, which removes each SNP one by one and evaluates its impact on the overall results. Notably, if the results change significantly after removing a particular SNP, this indicates that the SNP is sensitive. In this study, no SNPs with significant deviation were found, indicating that the analysis results are stable and reliable.

Given that genetic variations may influence multiple correlated exposures, multivariable Mendelian randomization (MVMR) was performed to delineate the direct causal effects of individual variable factors on MSMI. This approach isolates the independent contribution of each exposure to disease risk [29]. To mitigate potential collinearity among exposures, the least absolute shrinkage and selection operator (LASSO) was applied for variable selection and dimensionality reduction. This method effectively identified a subset of influential exposures by retaining factors significantly associated with MSMI while excluding those with negligible effects [30]. The optimal tuning parameter (λ) was determined using 10-fold cross-validation, selecting the value that minimized the mean cross-validated error. Variables with non-zero coefficients were retained for subsequent multivariable MR analysis.

## Results

### The global and regional burden of MSMI in 2021

Globally, the country-specific burden of MSMI in 2021 is detailed in Figs 2–3 and 5. Fig 2 presents the absolute numbers of incidence, prevalence, mortality, DALYs, YLDs, and YLLs for each country in 2021. The findings reveal that China, India, and Pakistan reported the highest number of incidence cases and prevalence cases, each exceeding 1,000,000 incidence cases and 150,000 prevalence cases. In terms of mortality attributable to MSMI, India, Nigeria, and the Democratic Republic of the Congo recorded the highest mortality, each experiencing over 1,500 annual deaths. Countries with elevated mortality rates were largely concentrated in Asia and Africa. India, Nigeria, and the Democratic Republic of the Congo also reported the highest number of DALY cases. Furthermore, the assessment of ASR metrics revealed that the highest values for ASIR, ASMR, ASDR, and ASPR in 2021 were predominantly observed in African and South American countries (Fig 3). Within Asia, Afghanistan and Pakistan shouldered the most substantial MSMI burden, whereas the ASR metrics of other Asian countries were generally comparable to those of Europe, North America, and Oceania. At the regional level, Western Sub-Saharan Africa was observed to have the highest number of mortality, DALYs and YLLs in 2021, followed by Sourth Asia, Eastern Sub-Saharan Africa and Central Latin America, whereas South Asia reported the highest number of YLDs, prevalence and incidence in 2021, followed by Western Sub-Saharan Africa, Eastern Sub-Saharan Africa and Central Latin America (Fig 5A). In contrast, six key metrics via age-standardized exhibited different regional trends. Specifically, Eastern Sub-Saharan Africa reported the highest age-standardized rates for mortality, DALYs and YLLs, Andean Latin America exhibited the highest age-standardized rates for YLDs and incidence, and Central Sub-Saharan Africa demonstrated the highest age-standardized rates for prevalence (Fig 5B).

Assessment of the MSMI burden across different age groups in 2021 (Fig 4) revealed a progressive decline in all ASR metrics with advancing age, with the notable exception of the 15–19 age group. The highest burden was observed in the 20–24 age group, whereas the lowest values for all burden metrics were recorded in the 45–49 age group. Furthermore,

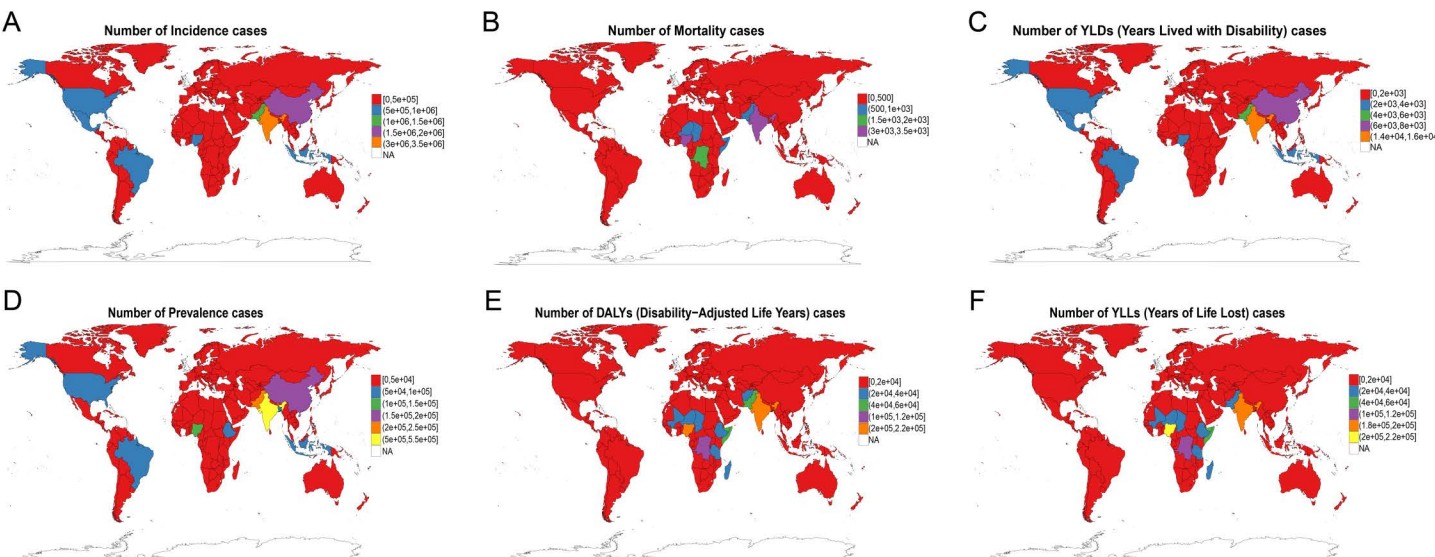

**Fig 2. Global Burden of MSMI in 2021.** Spatial distribution of MSMI case burden across 204 countries and territories is presented as follows: **(A)** Incidence cases, **(B)** Mortality cases, **(C)** YLDs, **(D)** Prevalence cases, **(E)** DALYs, and **(F)** YLLs. These maps visualize geographical disparities in disease impact, highlighting regions with the highest burden across different health metrics. YLDs, Years Lived with Disability; DALYs, Disability-Adjusted Life Years; YLLs, Years of Life Lost. The map layer (country/region boundaries) was obtained from Natural Earth (https://www.naturalearthdata.com/), which is in the public domain (Terms of use:https://www.naturalearthdata.com/about/terms-of-use/). The basemap was created using the "rnaturalearth" package in R software. Map boundaries are only for illustrative purposes and do not imply any political statement.

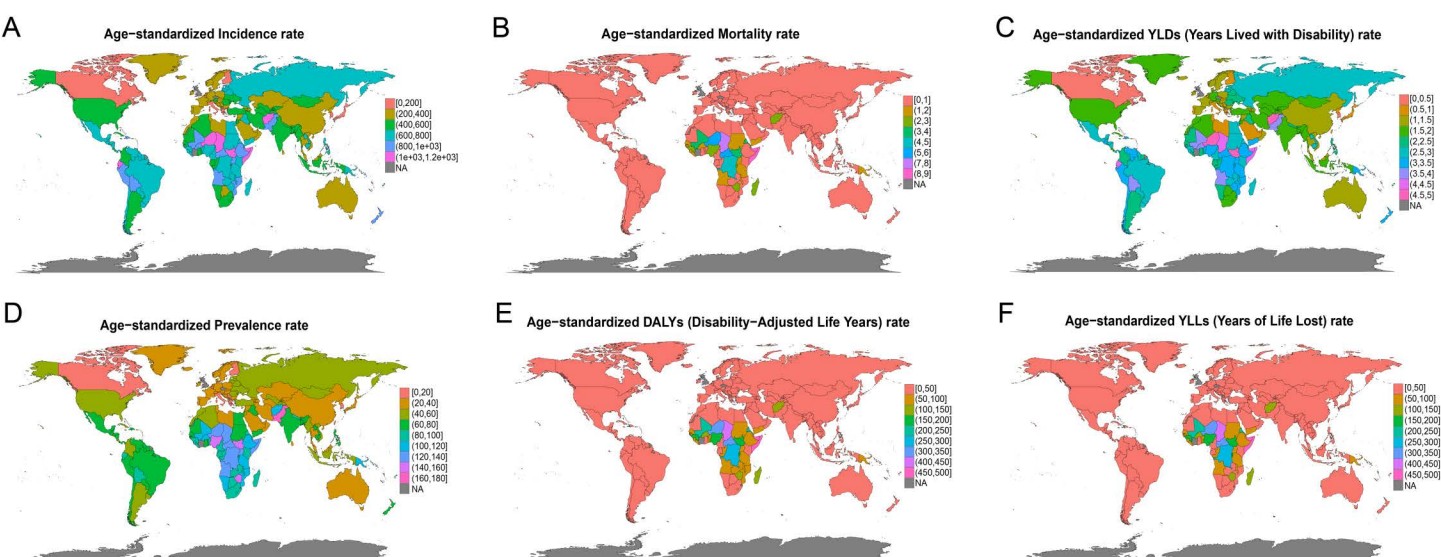

**Fig 3. Global Age-Standardized Burden of MSMI in 2021.** Geographic distribution of age-standardized MSMI burden metrics across 204 countries and territories: **(A)** Age-Standardized Incidence Rate (ASIR), **(B)** Age-Standardized Mortality Rate (ASMR), **(C)** Age-Standardized YLDs Rate (AS-YLD), **(D)** Age-Standardized Prevalence Rate (ASPR), **(E)** Age-Standardized DALYs Rate (ASDR), and **(F)** Age-Standardized YLLs Rate (AS-YLL). These maps enable cross-national comparisons of disease burden, independent of population age structure differences. YLDs, Years Lived with Disability; DALYs, Disability-Adjusted Life Years; YLLs, Years of Life Lost. The map layer (country/region boundaries) was obtained from Natural Earth (https://www.naturalearthdata.com/), which is in the public domain (Terms of use:https://www.naturalearthdata.com/about/terms-of-use/). The basemap was created using the "rnaturalearth" package in R software. Map boundaries are only for illustrative purposes and do not imply any political statement.



**Fig 4. Age-Specific Burden of MSMI in 2021. (A)** Age-stratified patterns of MSMI burden expressed as ASR, including Age-Standardized DALYs Rate, Age-Standardized Mortality Rate, Age-Standardized Incidence Rate, Age-Standardized Prevalence Rate, Age-Standardized YLDs Rate, and Age-Standardized YLLs Rate, highlighting variations in disease impact across the lifespan. **(B)** Absolute case numbers of MSMI burden, namely DALYs, Mortality, Incidence, Prevalence, YLDs and YLLs, across different age groups, representing the actual population-level disease distribution. DALYs, Disability-Adjusted Life Years; YLDs, Years Lived with Disability; YLLs, Years of Life Lost.

given that the economic impact of disease varies with socioeconomic development, populations were stratified by the SDI to evaluate the MSMI burden across different development strata. As shown in Fig 6, a decline in the SDI was associated with a gradual increase in the ASIR, ASPR, ASMR, ASDR, AS-YLD, and AS-YLL. Further Spearman correlation analysis showed that SDI was significantly negatively correlated with ASIR, ASPR, ASMR, ASDR, AS-YLD, and AS-YLL (all P<0.001), whether for 1990 (S1 Fig) or 2021(S2 Fig). Consequently, these results demonstrate a clear negative correlation between SDI and MSMI burden, indicating that the majority of the MSMI burden is concentrated in low-SDI and low-middle-SDI countries and territories.



**Fig 5. Regional Burden of MSMI in 2021. (A)** Absolute number of MSMI cases by region, representing the actual magnitude of disease burden across different populations. **(B)** Age-standardized burden rates of MSMI across geographic regions, facilitating direct comparison of disease intensity while accounting for population age structures. 21 GBD regions: Andean Latin America, Australasia, Caribbean, Central Asia, Central Europe, Central Latin America, Central Sub−Saharan Africa, East Asia, Eastern Europe, Eastern Sub−Saharan Africa, High−income Asia Pacific, High−income North America, North Africa and Middle East, Oceania, South Asia, Southeast Asia, Southern Latin America, Southern Sub−Saharan Africa, Tropical Latin America, Western Europe and Western Sub−Saharan Africa. DALYs, Disability-Adjusted Life Years; YLDs, Years Lived with Disability; YLLs, Years of Life Lost.

Decomposition analysis of changes in disease burden among MSMI revealed marked heterogeneity across SDI quintiles (Fig 6C), as well as similar trends across Cause (S3 Fig) and Sex (S4 Fig). All of the total change in MSMI burden, including incidence, prevalence, mortality, DALYs, YLDs and YLLs, was predominantly driven by epidemiological change and population growth (S3 and S4 Figs). For SDI quintiles, distinct patterns were observed for mortality and morbidity indicators (incidence) (Fig 6C). The total change in DALYs among MSMI was predominantly driven by epidemiological change and population growth in lower SDI settings, with opposing effects across the development spectrum. In the Low SDI quintile, population growth contributed substantially to increased DALYs, while epidemiological change exerted a modest negative effect. Conversely, in High SDI and High-middle SDI quintiles, epidemiological change was the dominant driver of DALY reduction, offsetting small increases attributable to aging. The net change (black dots) demonstrated a gradient from high burden in Low SDI to minimal change in High SDI settings. A similar pattern was observed for mortality



Fig 6. Burden of MSMI by SDI in 2021. (A) Age-standardized burden rates of MSMI across SDI categories, revealing gradients in disease intensity relative to socioeconomic development levels. (B) Absolute number of MSMI cases by SDI stratum, demonstrating the actual distribution of disease burden across populations with varying socioeconomic status. (C) Decomposition results of MSMI burden across SDI categories at the SDI components level.

Three SDI components include population structure (aging), epidemiological changes, and population size (population). DALYs, Disability-Adjusted Life Years; YLDs, Years Lived with Disability; YLLs, Years of Life Lost.

and YLLs. In Low SDI countries, mortality change was driven primarily by population growth with minimal epidemiological improvement. In contrast, High SDI quintiles experienced mortality reductions through favorable epidemiological change. Notably, aging contributed negligibly to mortality changes across most quintiles except Low SDI, where a marginal aging effect was observed. The decomposition of incidence revealed distinct dynamics. Low SDI countries exhibited the largest increase in incidence cases, driven overwhelmingly by population growth and partially offset by epidemiological improvement. This pattern was attenuated at higher SDI levels, where epidemiological change contributed more substantially to incidence reduction. Aging effects on incidence were minimal across all quintiles.

Across all disease burden metrics, three consistent patterns emerged: (1) population growth was the dominant driver of increased burden in Low SDI settings; (2) epidemiological change contributed to burden reduction across all SDI levels, with effect magnitude positively correlated with development status; and (3) aging exerted minimal influence on most indicators, with slight positive contributions to mortality, incidence and YLDs in lower SDI settings. The net change in disease burden (black dots) demonstrated a clear SDI gradient, with Low SDI countries experiencing the largest change in all metrics and High SDI countries approaching zero net change or reductions.

## Global trends in the burden of MSMI from 1990 to 2021

The global burden trends of MSMI from 1990 to 2021 is presented in Fig 7. The findings reveal a consistent overall decline in all ASR, including ASIR, ASPR, ASMR, AS-YLD, AS-YLL, and ASDR, over this period. By 2021, each of these ASR metrics showed a substantial decrease compared to its 1990 levels. A parallel reduction was observed in the absolute numbers of incidence, prevalence, mortality, YLLs, YLDs, and DALYs, which were all significantly lower in 2021 than in 1990.

As shown in Fig 8, the age distribution of MSMI burden from 1990 to 2021 revealed a general decline in ASR (ASIR, ASMR and ASDR) with advancing age, except in the 15–19 age group. The highest values for ASIR, ASMR, ASPR, and ASDR were consistently observed in the 20–24 age group, while all metrics remained at relatively low levels beyond age 40. Furthermore, compared with 1990, all ASR metrics in 2021showed significant reductions among individuals aged 15–29 years. In contrast, only minimal changes or gradual declines were observed for the 30–49 age group.

Despite consistent diagnostic criteria for MSMI across countries, disparities in socioeconomic development have led to incomplete uniformity in diagnostic quality, potentially affecting the accuracy of global burden assessments. To address this, health losses attributable to MSMI from 1990 to 2021 were further analyzed by SDI, as shown in Fig 9. The study found that, although all ASR indicators exhibited a gradual downward trend in low-SDI regions, the absolute numbers of incidence cases, prevalence cases, and YLDs rose significantly. Specifically, these indicators increased from 3,061,689 (95% UI 2,301,567-4,006,663), 353,651 (95% UI: 239,741–520,932), and 12,526 (95% UI: 5,919–22,393) in 1990– 4,711,030 (95% UI: 3,557,613–6,151,574), 597,803 (95% UI: 423,736–852,979), and 19,630 (95% UI: 9,178–35,332) in 2021, representing increases of 54%, 69%, and 57%, respectively. In contrast, mortality rate, YLLs, and DALYs remained largely unchanged in this region. Population growth was identified as the primary driver of this phenomenon; approximately 70%–75% of the increase in MSMI cases in low-SDI regions was attributable to population growth. Meanwhile, the burden of MSMI remained consistently low in high-SDI and high-middle-SDI regions over the study period.

National-level trends in the MSMI burden from 1990 to 2021 revealed divergent patterns across countries, as shown in Fig 10. Globally, the ASIR of MSMI declined in most countries during this period, with the exception of ten nations, including Australia, Belarus, Georgia, Germany, Latvia, Kazakhstan, Romania, Russia, Samoa and Spain. Saudi Arabia demonstrated the most substantial decrease in ASIR. A similar overall downward trend was observed for ASMR worldwide.



**Fig 7. Temporal Trends in MSMI Burden from 1990 to 2021. (A)** Mortality counts and ASMR. **(B)** DALYs and ASDR. **(C)** YLDs and AS-YLD. **(D)** YLLs and AS-YLL. **(E)** Prevalence counts and ASPR. **(F)** Incidence counts and ASIR. The bar chart represents the absolute cases, and the dotted line chart stands for age-standardized burden rates. DALYs, Disability-Adjusted Life Years; YLDs, Years Lived with Disability; YLLs, Years of Life Lost. ASMR, Age-Standardized Mortality Rate; ASDR, Age-Standardized DALYs Rate; AS-YLD, Age-Standardized YLDs Rate; AS-YLL, Age-Standardized YLLs Rate; ASIR, Age-Standardized Incidence Rate; ASPR, Age-Standardized Prevalence Rate.

However, Kazakhstan experienced a significant increase in ASMR, while Canada, the Democratic Republic of the Congo, Zimbabwe, Thailand, and Papua New Guinea exhibited more modest elevations. Furthermore, assessment of trends in the ASDR revealed that all countries and regions displayed a decline in DALYs, with the exception of five countries, including Australia and Kazakhstan. The most substantial reductions in ASDR were observed in Asian and African nations.

**Future trends in the burden of MSMI**

To forecast future trends in the burden of MSMI, both the ARIMA and BAPC models were employed. Projections from the ARIMA model suggest that ASPR and ASDR of MSMI will continue to decline in the coming years, while the ASIR is projected to rise, and the ASMR is expected to remain stable (Fig 11). Specifically, by 2050, ASPR and ASDR are estimated to decrease from 60.20 to 55.11 and from 28.57 to 7.74, respectively, whereas ASIR is projected to increase from 492.69 to 887.40. In contrast, projections from the BAPC model differ slightly from those of the ARIMA model. All four metrics, including ASIR, ASPR, ASMR, and ASDR, exhibited a consistent decline through 2041, with the magnitude of reduction illustrated by the blue lines (Fig 12), Based on these findings,it is concluded that ASPR and ASDR are likely to continue

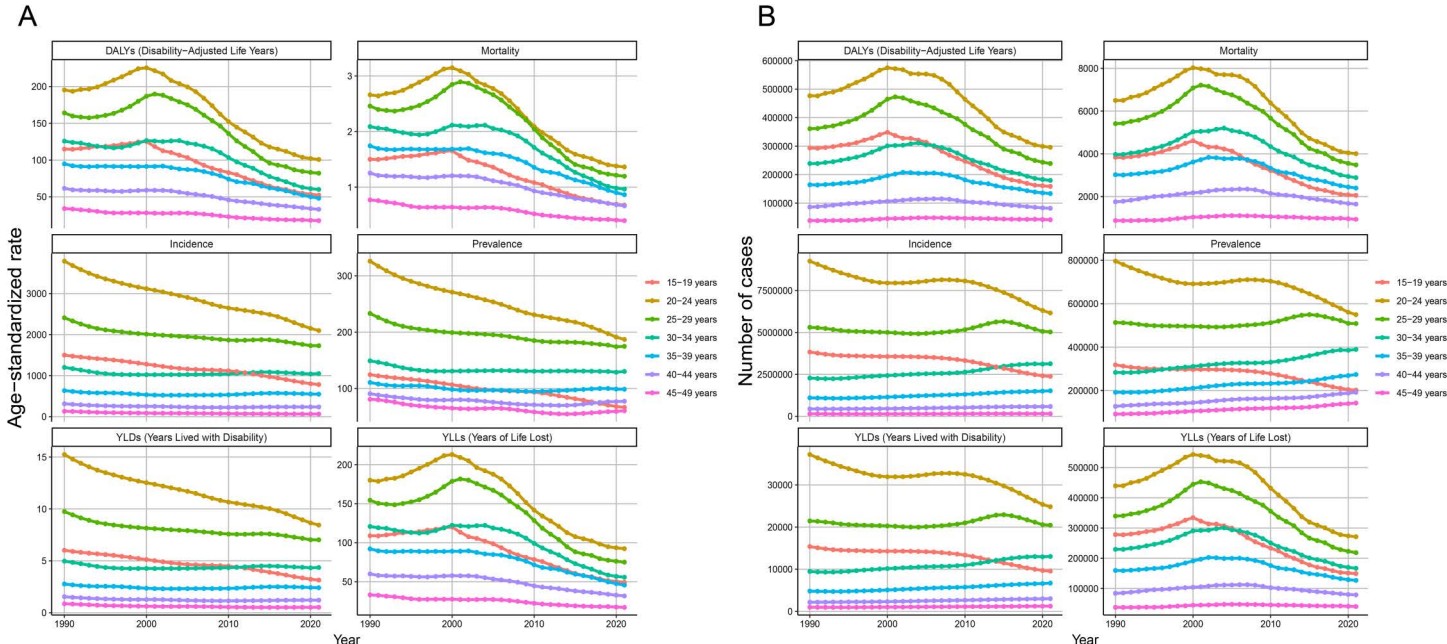

**Fig 8. Age-Stratified Burden Trends of MSMI from 1990 to 2021. (A)** Temporal patterns in age-standardized burden rates across age groups, showing how disease intensity has evolved over three decades while controlling for demographic changes. **(B)** Longitudinal trends in absolute case numbers by age group, revealing shifts in the actual population distribution of MSMI burden across different generations. DALYs, Disability-Adjusted Life Years; YLDs, Years Lived with Disability; YLLs, Years of Life Lost.

decreasing in the future. Nevertheless, the trajectory of ASIR remain uncertain, and ASMR is expected to either remain stable or undergo a slow, sustained decline.

## Two-sample MR analysis and multivariable MR analysis

To investigate potential causal links, a range of variable factors were systematically evaluated in relation to MSMI. For maternal sepsis, multiple factors showed statistically significant associations ($P<0.05$), as detailed in Table 2. These included lipid profiles, inflammatory markers, medication use, and pregnancy history variables. Using the inverse variance weighted (IVW) method, we identified several factors with odds ratios (OR) > 1, such as LDL cholesterol, previous smoking, RANTES levels, IL-13 levels, IL-10 levels, birth weight, cholesterol-lowering medications, blood pressure medication, and PSG11, indicating their potential roles as risk factors for the development of maternal sepsis. Conversely, those factor with OR < 1, including HDL cholesterol, TG, TC, IL-1Ra levels, IFN-γ levels, PSG9, and number of pregnancy terminations, emerged as potential protective influences against the development of maternal sepsis.

Analysis of other maternal infections revealed distinct causal profiles, as detailed in Table 3. Statistically significant causal relationships ($P<0.05$) were identified between multiple factors and other maternal infections, including spontaneous miscarriage or termination, BMI, CRP, LDL cholesterol, TC, TG, IL-13 levels, IL-2Ra levels, NT-proBNP, cholesterol-lowering medications, blood pressure medication, and vitamin D levels. These findings indicated their potential causal roles in the development of other maternal infections. Using the IVW method, factors with OR > 1 were classified as potential risk factors of other maternal infections, such as spontaneous miscarriage or termination, CRP, IL-13 levels, NT-proBNP, blood pressure medication, and vitamin D levels. Conversely, the IVW method found that BMI, LDL cholesterol, total cholesterol, triglycerides, IL-2Ra levels, and cholesterol-lowering medication (all with OR < 1) emerged as potential protective factors, suggesting a negative association with infection risk.

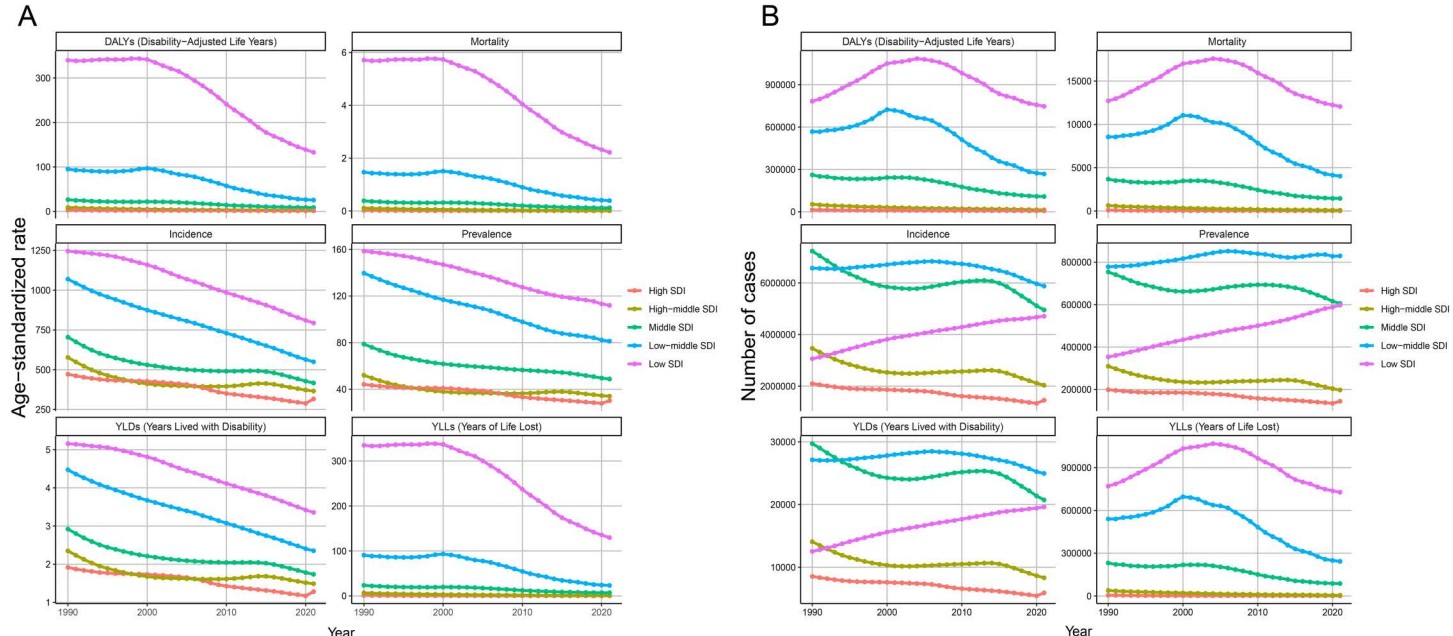

**Fig 9. Socioeconomic Disparities in MSMI Burden from 1990 to 2021. (A)** Trends in age-standardized burden rates across SDI categories, illustrating the evolution of disease intensity in relation to socioeconomic development over three decades. **(B)** Temporal patterns in absolute case numbers by SDI stratum, revealing how the actual distribution of MSMI burden has shifted across populations with varying socioeconomic status. DALYs, Disability-Adjusted Life Years; YLDs, Years Lived with Disability; YLLs, Years of Life Lost.

All instrumental variables showed F-statistics greater than 10 (S7 Table), indicating a low risk of weak instrument bias. The results of harmonization analyses in the Mendelian randomization analysis were available in S8 Table. The results of causal effects using five Mendelian randomization methods were summarized in S9 Table. The validity of MR estimates was systematically evaluated through pleiotropy (S10 Table), heterogeneity (S11 Table), and directionality tests (S12 Table). As summarized in Table 4, all Cochran's Q statistics returned P-values greater than 0.05, indicating negligible heterogeneity among instrumental variables and supporting the use of fixed-effects IVW models. Similarly, pleiotropy assessment via MR-Egger and MR-PRESSO showed no evidence of horizontal pleiotropy (all P > 0.05, Table 5), confirming the robustness of causal estimates. The detailed results of MR-PRESSO global test were available in S13 Table. Finally, directionality testing verified the assumed causal direction, with outcome variance consistently lower than exposure variance across all analyses (Table 6).The results of the sensitivity analysis showed that no significantly deviating SNPs were found in this study (S5 Fig for maternal sepsis and S6 Fig for other maternal infections), indicating that the analysis results are stable and reliable.

LASSO regression retained key inflammatory biomarkers, including RANTES, IL-10, CRP, IL-13, and NT-proBNP, for subsequent multivariable MR analysis (MVMR). MVMR analysis was employed to evaluate the independent causal roles of specific inflammatory biomarkers in MSMI. For maternal sepsis, genetically predicted elevations in both RANTES and IL-10 levels demonstrated significant positive associations with disease risk (Table 7). Relatively, in the analysis of other maternal infections, higher levels of CRP, IL-13, and NT-proBNP were causally linked to increased infection risk (Table 7). These results highlight distinct inflammatory pathways contributing to different forms of maternal infection and underscore the potential value of these biomarkers in risk stratification.



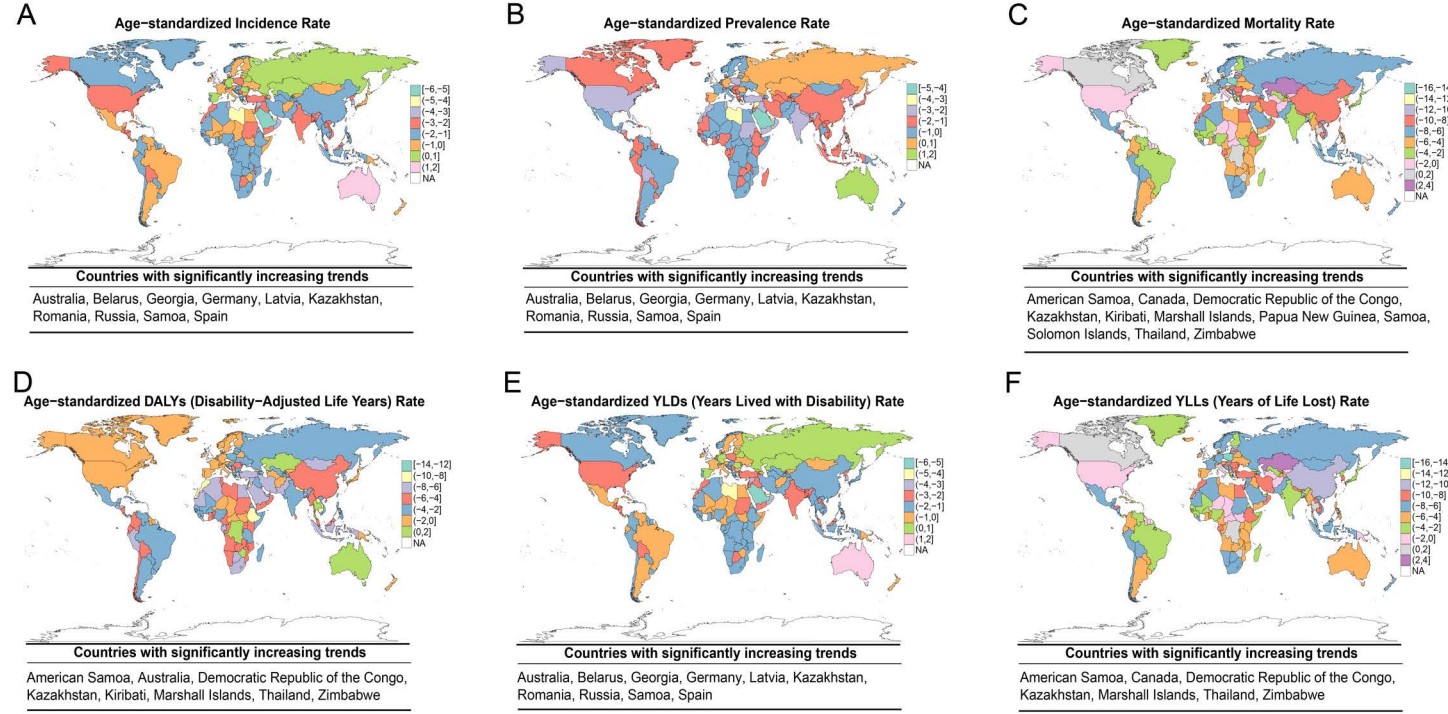

**Fig 10. Global Trends in Age-Standardized Burden of MSMI from 1990 to 2021.** Spatiotemporal patterns of EAPC across 204 countries and territories are shown for: **(A)** ASIR, **(B)** ASPR, **(C)** ASMR, **(D)** ASDR, **(E)** AS-YLD, and **(F)** AS-YLL. Countries with significantly increasing trends were highlighted below each small chart. These maps visualize geographical heterogeneity in long-term trends, highlighting regions with accelerating or decelerating disease burden. ASIR, Age-Standardized Incidence Rate; ASPR, Age-Standardized Prevalence Rate; ASMR, Age-Standardized Mortality Rate; ASDR, Age-Standardized DALYs Rate; AS-YLD, Age-Standardized YLDs Rate; AS-YLL, Age-Standardized YLLs Rate. The map layer (country/region boundaries) was obtained from Natural Earth (https://www.naturalearthdata.com/), which is in the public domain (Terms of use:https://www.naturale-arthdata.com/about/terms-of-use/). The basemap was created using the "rnaturalearth" package in R software. Map boundaries are only for illustrative purposes and do not imply any political statement.

## Discussion

This comprehensive study firstly provides a detailed assessment of the global burden, trends, and causal determinants of MSMI from 1990 to 2021, with projections extending to 2041. Our findings reveal persistent disparities, identify modifiable risk factors, and highlight critical areas for future intervention and research.

The disproportionate burden in low- and middle-income countries, particularly in South Asia and sub-Saharan Africa, aligns with the inverse equity hypothesis proposed by Victora et al. [31], which posits that health interventions often benefit higher-resource populations first, thereby widening disparities. For example, increasing mortality was observed in Kazakhstan, a phenomenon that may be related to limited healthcare access and uneven resource distribution. In contrast, a marked decline was demonstrated in Saudi Arabia, likely reflecting improvements in maternal healthcare services. Additionally, increasing trends observed in Australia may be associated with advanced maternal age and a higher prevalence of comorbidities. The strong negative correlation between SDI and MSMI burden (Fig 6) underscores how structural determinants, including limited healthcare access, sanitation, and education-perpetuate inequitable maternal health outcomes [32]. Moreover, as a composite indicator, the SDI encompasses components such as educational attainment and fertility rate, which may also influence the burden of infections among populations with MSMI [33–34]. Notably, with respect to fertility, primiparous women tend to face greater difficulties in newborn care compared with multiparous women [35]. The observed age distribution pattern, characterized by a peak burden in the 20–24 age group and elevated risk



Fig 11. **Projected Trends in MSMI Burden from 2022 to 2050 using ARIMA Model. (A)** Projected trends in incidence counts and ASIR. **(B)** Projected trends in prevalence counts and ASPR. **(C)** Projected trends in mortality counts and ASMR. **(D)** Projected trends in DALYs and ASDR. ARIMA, autoregressive integrated moving average model. The bar chart represents the absolute cases, and the dotted line chart stands for age-standardized burden rates. DALYs, Disability-Adjusted Life Years; ASIR, Age-Standardized Incidence Rate; ASPR, Age-Standardized Prevalence Rate; ASMR, Age-Standardized Mortality Rate; ASDR, Age-Standardized DALYs Rate.

in adolescents (15–19 years), reflects complex biological and social vulnerabilities. Younger mothers face increased physiological risks due to reproductive immaturity while simultaneously experiencing greater barriers to healthcare utilization [36]. Thus, this pattern emphasizes the need for life-course approaches to maternal health that address specific risk factors across different age groups. The overall global decline in MSMI burden since 1990 likely reflects improvements in maternal healthcare, antibiotic accessibility, and infection control practices [37]. Nevertheless, divergent trends between high- and low-SDI regions (Fig 9)-where absolute case numbers increased in low-SDI areas despite declining

**A**

**Global Female Incidence**



**B**

**Global Female Prevalence**

**C**

**Global Female Mortality**

**D**

**Global Female DALYs**

**Fig 12. Trend in the burden of MSMI from 2022 to 2041 using BAPC model. (A)** Trend of incidence and ASIR from 2022 to 2041. **(B)** Trend of prevalence and ASPR from 2022 to 2041. **(C)** Trend of mortality and ASMR from 2022 to 2041. **(D)** Trend of DALYs and ASDR from 2022 to 2050. The blue line represents the predicted trend, and the grey-shaded area represents the 95% confidence interval of the predicted values; the grey-dotted vertical line divides the data into real values (1990-2021) and predicted values (2022-2041). BAPC model, Bayesian age-period-cohort.

ASR-suggest that population growth is the main driver of the increasing absolute burden, offsetting the decline in ASR, and highlights how persistent structural barriers can undermine progress. Consequently, this finding supports the concept of the epidemiological polarization described by Bawah et al [38], where improvements in health indicators mask persistent or worsening absolute burdens in disadvantaged populations.

The MR analyses provide novel causal evidence linking specific biomarkers to MSMI risk. The identification of inflammatory markers (RANTES, IL-10, CRP) as risk factors reinforces the central role of dysregulated immunity in sepsis pathogenesis [39]. Importantly, the differential associations observed in the MR analysis may reflect distinct immunopathological mechanisms underlying MSMI. RANTES and IL-10, identified as independent risk factors for maternal sepsis, are closely involved in immune regulation and anti-inflammatory processes. Elevated IL-10 levels, in particular, may indicate an excessive compensatory anti-inflammatory response, leading to immune suppression and impaired pathogen clearance during sepsis progression [40]. Similarly, RANTES plays a key role in leukocyte recruitment and immune cell activation, and its dysregulation may contribute to maladaptive immune responses in severe infections [41]. In contrast,

**Table 2. Results of the relationship between the variable factors and the risk of maternal sepsis.**

| id.exposure | Exposure | SNPs | Method | P value | OR | Lower 95%CI | Upper 95%CI |
|---|---|---|---|---|---|---|---|
| ieu-a-27 | Birth weight | 13 | MR Egger | 0.026189844 | 17.61498355 | 1.970634157 | 157.4557329 |
| | | | Weighted median | 0.014247002 | 1.837887361 | 1.129660277 | 2.990129 |
| | | | Inverse variance weighted | 0.014684896 | 1.653291065 | 1.103972225 | 2.475942133 |
| | | | Simple mode | 0.081865576 | 2.246910894 | 0.974331178 | 5.181614508 |
| | | | Weighted mode | 0.069196648 | 2.035202449 | 1.012728669 | 4.089988895 |
| ukb-a-224 | Smoking status: Previous | 151 | MR Egger | 0.051806864 | 51.07375485 | 1.000928611 | 2606.108373 |
| | | | Weighted median | 0.002474608 | 4.336763944 | 1.676947379 | 11.21533194 |
| | | | Inverse variance weighted | 0.017712179 | 2.176941596 | 1.14454347 | 4.140580797 |
| | | | Simple mode | 0.175262762 | 8.157792703 | 0.39782714 | 167.2826589 |
| | | | Weighted mode | 0.149442164 | 11.34477924 | 0.424578392 | 303.1336929 |
| ukb-a-448 | Medication for choles-terol blood pressure dia-betes or take exogenous hormones: Cholesterol lowering medication | 10 | MR Egger | 0.169756592 | 19.87727453 | 0.40912769 | 965.7279435 |
| | | | Weighted median | 0.02758256 | 6.095166926 | 1.220792874 | 30.43191081 |
| | | | Inverse variance weighted | 0.00486639 | 6.209318177 | 1.741878343 | 22.13451494 |
| | | | Simple mode | 0.263178546 | 4.878170969 | 0.361405876 | 65.84439704 |
| | | | Weighted mode | 0.137482275 | 6.53350567 | 0.684122105 | 62.39631204 |
| ukb-b-18009 | Medication for choles-terol, blood pressure, diabetes, or take exog-enous hormones: Blood pressure medication | 20 | MR Egger | 0.748570165 | 2.801366397 | 0.005667237 | 1384.740787 |
| | | | Weighted median | 0.149057338 | 3.037928239 | 0.671493829 | 13.74399525 |
| | | | Inverse variance weighted | 0.038760688 | 3.432065962 | 1.065747486 | 11.05240868 |
| | | | Simple mode | 0.327438317 | 4.077770878 | 0.26312739 | 63.19454362 |
| | | | Weighted mode | 0.413179614 | 2.861438751 | 0.243767897 | 33.5886383 |
| ukb-a-355 | Number of pregnancy terminations | 28 | MR Egger | 0.287195296 | 0.419467804 | 0.087520754 | 2.010417297 |
| | | | Weighted median | 0.233074067 | 0.61861041 | 0.280920143 | 1.362233534 |
| | | | Inverse variance weighted | 0.0210607 | 0.504499724 | 0.282100298 | 0.902232198 |
| | | | Simple mode | 0.85021559 | 1.154661131 | 0.26328662 | 5.063843838 |
| | | | Weighted mode | 0.967277967 | 1.026820841 | 0.29333479 | 3.594394788 |
| ebi-a-GCST000755 | HDL cholesterol | 130 | MR Egger | 0.065685167 | 0.8253907 | 0.674018292 | 1.010758633 |
| | | | Weighted median | 0.516823887 | 0.949153684 | 0.81061175 | 1.111373868 |
| | | | Inverse variance weighted | 0.005521824 | 0.882204584 | 0.807464098 | 0.963863198 |
| | | | Simple mode | 0.039523617 | 0.605877579 | 0.377837325 | 0.971549435 |
| | | | Weighted mode | 0.977230465 | 1.003004879 | 0.81656924 | 1.232006714 |
| ebi-a-GCST000758 | Triglycerides | 143 | MR Egger | 0.009163818 | 0.771160325 | 0.63596414 | 0.935097137 |
| | | | Weighted median | 0.000143682 | 0.80930529 | 0.72567046 | 0.902579186 |
| | | | Inverse variance weighted | 0.000008693679 | 0.859187247 | 0.796469044 | 0.926844215 |
| | | | Simple mode | 0.023429635 | 0.712645161 | 0.533345432 | 0.952221758 |
| | | | Weighted mode | 0.016152045 | 0.74230209 | 0.58396573 | 0.943569742 |
| ieu-a-783 | Triglycerides | 165 | MR Egger | 0.026624249 | 0.811708821 | 0.676127916 | 0.974477159 |
| | | | Weighted median | 0.006503945 | 0.868133426 | 0.784066002 | 0.961214544 |
| | | | Inverse variance weighted | 0.00012649 | 0.87085792 | 0.811411194 | 0.934659915 |
| | | | Simple mode | 0.024788914 | 0.707683813 | 0.524721496 | 0.954442278 |
| | | | Weighted mode | 0.04316658 | 0.712038099 | 0.513623233 | 0.987101482 |
| ieu-b-4850 | Triglycerides | 164 | MR Egger | 0.514636128 | 0.950092268 | 0.814772503 | 1.10788633 |
| | | | Weighted median | 0.119103355 | 0.932661869 | 0.854377866 | 1.018118793 |
| | | | Inverse variance weighted | 0.005936672 | 0.914919292 | 0.858760057 | 0.974751101 |
| | | | Simple mode | 0.874969231 | 1.019541065 | 0.801451326 | 1.296977059 |
| | | | Weighted mode | 0.677463558 | 1.040494333 | 0.863268529 | 1.254103932 |

*(Continued)*

**Table 2.** (Continued)

| id.exposure | Exposure | SNPs | Method | P value | OR | Lower 95%CI | Upper 95%CI |
|---|---|---|---|---|---|---|---|
| ebi-a-GCST005068 | LDL cholesterol | 10 | MR Egger | 0.542652878 | 1.105400094 | 0.811641127 | 1.505479857 |
| | | | Weighted median | 0.067559961 | 1.174462584 | 0.988448546 | 1.395482211 |
| | | | Inverse variance weighted | 0.004129546 | 1.222608152 | 1.065704536 | 1.402612677 |
| | | | Simple mode | 0.282342085 | 1.156885747 | 0.9011695 | 1.485164146 |
| | | | Weighted mode | 0.222616731 | 1.166143322 | 0.92658473 | 1.467637232 |
| ebi-a-GCST90101747 | Total cholesterol levels | 8 | MR Egger | 0.35339293 | 0.788616958 | 0.496436891 | 1.25276086 |
| | | | Weighted median | 0.03148423 | 0.727618402 | 0.544582631 | 0.972173016 |
| | | | Inverse variance weighted | 0.01640527 | 0.753491828 | 0.597975198 | 0.949453986 |
| | | | Simple mode | 0.08750503 | 0.672150117 | 0.454072204 | 0.994964624 |
| | | | Weighted mode | 0.060587159 | 0.715926922 | 0.534015199 | 0.959806684 |
| ebi-a-GCST004431 | RANTES levels | 121 | MR Egger | 0.311158283 | 1.065319734 | 0.943031004 | 1.203466409 |
| | | | Weighted median | 0.00018884 | 1.140730975 | 1.064543343 | 1.222371232 |
| | | | Inverse variance weighted | 0.000002851483 | 1.125730823 | 1.071270839 | 1.182959378 |
| | | | Simple mode | 0.068437547 | 1.222763384 | 0.986815698 | 1.515126175 |
| | | | Weighted mode | 0.068815772 | 1.207212398 | 0.987374804 | 1.47599652 |
| ebi-a-GCST004443 | Interleukin-13 levels | 139 | MR Egger | 0.471625071 | 1.035113669 | 0.942521091 | 1.136802474 |
| | | | Weighted median | 0.108217887 | 1.061069258 | 0.987030259 | 1.140662062 |
| | | | Inverse variance weighted | 0.039725047 | 1.045933268 | 1.00211159 | 1.091671238 |
| | | | Simple mode | 0.842634635 | 1.020647571 | 0.834471163 | 1.248361249 |
| | | | Weighted mode | 0.040195098 | 1.119939023 | 1.006104332 | 1.246653429 |
| ebi-a-GCST004444 | Interleukin-10 levels | 7 | MR Egger | 0.193901758 | 1.254133972 | 0.932914526 | 1.685955119 |
| | | | Weighted median | 0.004550546 | 1.258570964 | 1.073695067 | 1.475280013 |
| | | | Inverse variance weighted | 0.002377289 | 1.229558639 | 1.076110241 | 1.404888077 |
| | | | Simple mode | 0.057193262 | 1.331528912 | 1.048481215 | 1.690988085 |
| | | | Weighted mode | 0.024490583 | 1.270998084 | 1.085806519 | 1.48777531 |
| ebi-a-GCST004447 | Interleukin-1-receptor antagonist levels | 105 | MR Egger | 0.025021545 | 0.864016141 | 0.761756259 | 0.980003619 |
| | | | Weighted median | 0.075208055 | 0.925423377 | 0.849689706 | 1.007907264 |
| | | | Inverse variance weighted | 0.035943042 | 0.93825573 | 0.884011244 | 0.995828752 |
| | | | Simple mode | 0.484748145 | 0.917567495 | 0.721443909 | 1.167007022 |
| | | | Weighted mode | 0.388876835 | 0.907349869 | 0.727993942 | 1.130893731 |
| ebi-a-GCST004456 | Interferon gamma levels | 121 | MR Egger | 0.74627903 | 0.975794177 | 0.841476345 | 1.131552042 |
| | | | Weighted median | 0.001997875 | 0.840715698 | 0.753116693 | 0.938503809 |
| | | | Inverse variance weighted | 0.000132407 | 0.868800803 | 0.808344967 | 0.93377811 |
| | | | Simple mode | 0.154090401 | 0.81715626 | 0.620108128 | 1.076819225 |
| | | | Weighted mode | 0.127668933 | 0.829193966 | 0.652710689 | 1.053395701 |
| prot-a-2400 | Pregnancy-specific beta-1-glycoprotein 11 | 20 | MR Egger | 0.619351044 | 1.071932022 | 0.818832584 | 1.403263966 |
| | | | Weighted median | 0.011313969 | 1.245484169 | 1.050902529 | 1.476093902 |
| | | | Inverse variance weighted | 0.000281446 | 1.261021025 | 1.112666064 | 1.429156579 |
| | | | Simple mode | 0.352150823 | 1.146872164 | 0.865403224 | 1.519887752 |
| | | | Weighted mode | 0.119983579 | 1.237217813 | 0.9575263 | 1.598606655 |
| prot-a-2408 | Pregnancy-specific beta-1-glycoprotein 9 | 33 | MR Egger | 0.527536053 | 0.880872417 | 0.59694879 | 1.299837154 |
| | | | Weighted median | 0.012263072 | 0.872188274 | 0.783667809 | 0.970707711 |
| | | | Inverse variance weighted | 0.0000103725911 | 0.837499446 | 0.774015506 | 0.906190271 |
| | | | Simple mode | 0.004113414 | 0.719683357 | 0.584185695 | 0.886608725 |
| | | | Weighted mode | 0.266778577 | 0.908061182 | 0.768210447 | 1.073371382 |

SNPs, single nucleotide polymorphisms; OR, Odds ratio; 95%CI, 95% Confidence Interval.

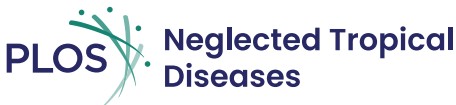

**Table 3. Results of the relationship between the variable factors and the risk of other maternal infection.**

| id.exposure | Exposure | SNPs | Method | P value | OR | Lower 95%CI | Upper 95%CI |
|---|---|---|---|---|---|---|---|
| ebi-a-GCST90095034 | Body mass index | 51 | MR Egger | 0.891030985 | 0.915329861 | 0.259856995 | 3.224191654 |
| | | | Weighted median | 0.512392841 | 0.864028786 | 0.557996018 | 1.337905147 |
| | | | Inverse variance weighted | 0.030158307 | 0.709525287 | 0.520276669 | 0.967612356 |
| | | | Simple mode | 0.936353471 | 1.034170713 | 0.455224881 | 2.34940819 |
| | | | Weighted mode | 0.793326083 | 0.925320026 | 0.519354171 | 1.648618993 |
| ieu-a-94 | Body mass index | 67 | MR Egger | 0.89395571 | 1.08006344 | 0.349579277 | 3.336974216 |
| | | | Weighted median | 0.52106033 | 0.875543194 | 0.583410457 | 1.313956367 |
| | | | Inverse variance weighted | 0.015545607 | 0.717571317 | 0.548402385 | 0.938924792 |
| | | | Simple mode | 0.004967441 | 0.243077147 | 0.093661257 | 0.630853154 |
| | | | Weighted mode | 0.906160939 | 0.961885166 | 0.505350699 | 1.830853454 |
| ieu-b-4815 | Body mass index | 59 | MR Egger | 0.044589208 | 0.826465969 | 0.689019628 | 0.991330248 |
| | | | Weighted median | 0.258357482 | 0.95324337 | 0.877286591 | 1.0357766 |
| | | | Inverse variance weighted | 0.000997478 | 0.908197398 | 0.857582139 | 0.961800016 |
| | | | Simple mode | 0.577389309 | 1.056845939 | 0.871014527 | 1.282324581 |
| | | | Weighted mode | 0.979885109 | 0.997981666 | 0.853504959 | 1.166914609 |
| ukb-b-12621 | Ever had stillbirth, spontaneous miscarriage or termination | 215 | MR Egger | 0.01644041 | 16.68877445 | 1.704534707 | 163.3966099 |
| | | | Weighted median | 0.544530301 | 1.651096394 | 0.326136004 | 8.358841922 |
| | | | Inverse variance weighted | 0.023127631 | 3.359175266 | 1.180669579 | 9.557338195 |
| | | | Simple mode | 0.413928906 | 0.132250775 | 0.00104164 | 16.79108411 |
| | | | Weighted mode | 0.51919405 | 0.226339319 | 0.002488983 | 20.5824982 |
| ukb-a-448 | Medication for cholesterol, blood pressure, diabetes, or take exogenous hormones: Cholesterol lowering medication | 10 | MR Egger | 0.390635006 | 0.063640621 | 0.000166036 | 24.3931167 |
| | | | Weighted median | 0.017017921 | 0.047486031 | 0.003886562 | 0.580184555 |
| | | | Inverse variance weighted | 0.008457575 | 0.073335426 | 0.010488928 | 0.5127392 |
| | | | Simple mode | 0.12270679 | 0.02929597 | 0.000504179 | 1.702280609 |
| | | | Weighted mode | 0.124975226 | 0.043023747 | 0.001123678 | 1.647307126 |
| ukb-b-17805 | Medication for cholesterol, blood pressure, diabetes, or take exogenous hormones: Cholesterol lowering medication | 28 | MR Egger | 0.03210947 | 0.011029112 | 0.000223011 | 0.545449518 |
| | | | Weighted median | 0.017312032 | 0.074679913 | 0.008815745 | 0.632628247 |
| | | | Inverse variance weighted | 0.002837364 | 0.101992556 | 0.022779337 | 0.456663046 |
| | | | Simple mode | 0.070572233 | 0.02811325 | 0.000682422 | 1.158161477 |
| | | | Weighted mode | 0.11964277 | 0.069290551 | 0.002672177 | 1.796729965 |
| ukb-b-18009 | Medication for cholesterol, blood pressure, diabetes, or take exogenous hormones: Blood pressure medication | 20 | MR Egger | 0.882245411 | 0.482545482 | 0.000035898 | 6486.521616 |
| | | | Weighted median | 0.060928905 | 10.40175695 | 0.898111116 | 120.4712266 |
| | | | Inverse variance weighted | 0.009595695 | 10.71577348 | 1.780601582 | 64.4882058 |
| | | | Simple mode | 0.101129173 | 37.18033426 | 0.608107904 | 2273.243359 |
| | | | Weighted mode | 0.111957585 | 29.21290536 | 0.552304716 | 1545.150377 |
| ebi-a-GCST005067 | C-reactive protein levels | 17 | MR Egger | 0.796189563 | 1.124476056 | 0.468952759 | 2.696319355 |
| | | | Weighted median | 0.010931496 | 1.357544545 | 1.072779351 | 1.717899575 |
| | | | Inverse variance weighted | 0.00502943 | 1.301375771 | 1.082599023 | 1.564363964 |
| | | | Simple mode | 0.131685432 | 1.431396876 | 0.919601533 | 2.228026971 |
| | | | Weighted mode | 0.111779545 | 1.431396876 | 0.942682966 | 2.173474107 |
| ebi-a-GCST90018730 | C-reactive protein | 53 | MR Egger | 0.375498015 | 1.316761217 | 0.720300244 | 2.407135243 |
| | | | Weighted median | 0.018942073 | 1.569382231 | 1.077087337 | 2.28668605 |
| | | | Inverse variance weighted | 0.000706729 | 1.571755344 | 1.209870209 | 2.041884197 |
| | | | Simple mode | 0.305283359 | 1.362239862 | 0.758765945 | 2.445678349 |
| | | | Weighted mode | 0.082079074 | 1.515821229 | 0.957086853 | 2.400737184 |

*(Continued)*

**Table 3.** (Continued)

| id.exposure | Exposure | SNPs | Method | P value | OR | Lower 95%CI | Upper 95%CI |
|---|---|---|---|---|---|---|---|
| ebi-a-GCST000759 | LDL cholesterol | 166 | MR Egger | 0.437452869 | 0.905822499 | 0.706120345 | 1.162003624 |
| | | | Weighted median | 0.00448905 | 0.77493768 | 0.649961703 | 0.923944294 |
| | | | Inverse variance weighted | 0.0000026794 | 0.766586593 | 0.686055225 | 0.856570992 |
| | | | Simple mode | 0.095937475 | 0.706177982 | 0.469961911 | 1.061122892 |
| | | | Weighted mode | 0.263659628 | 0.846616324 | 0.63287721 | 1.132540704 |
| ebi-a-GCST005068 | LDL cholesterol | 10 | MR Egger | 0.093097307 | 0.631242684 | 0.393311362 | 1.013109118 |
| | | | Weighted median | 0.017423333 | 0.732439333 | 0.566626777 | 0.946773781 |
| | | | Inverse variance weighted | 0.003830062 | 0.733211185 | 0.594135036 | 0.904842517 |
| | | | Simple mode | 0.111775718 | 0.703794215 | 0.476235031 | 1.04008791 |
| | | | Weighted mode | 0.088567329 | 0.716807367 | 0.509285883 | 1.008888756 |
| ebi-a-GCST90018741 | LDL cholesterol | 85 | MR Egger | 0.321153962 | 0.88310166 | 0.691807659 | 1.127290991 |
| | | | Weighted median | 0.03014286 | 0.755764737 | 0.586747845 | 0.973468148 |
| | | | Inverse variance weighted | 0.000857192 | 0.770547508 | 0.66106398 | 0.898163384 |
| | | | Simple mode | 0.000345903 | 0.443534541 | 0.289361235 | 0.679852258 |
| | | | Weighted mode | 0.022201985 | 0.764203947 | 0.609491006 | 0.958189156 |
| ieu-a-781 | LDL cholesterol | 133 | MR Egger | 0.002382081 | 0.726461582 | 0.593492795 | 0.889221292 |
| | | | Weighted median | 0.00842772 | 0.784753714 | 0.655263274 | 0.939833524 |
| | | | Inverse variance weighted | 0.00001852412 | 0.78594139 | 0.703893288 | 0.877553287 |
| | | | Simple mode | 0.298023451 | 0.825466716 | 0.576012649 | 1.182951974 |
| | | | Weighted mode | 0.045221485 | 0.797397264 | 0.640255824 | 0.993106776 |
| ieu-b-4845 | LDL cholesterol | 66 | MR Egger | 0.029327673 | 0.753041996 | 0.586823025 | 0.966342873 |
| | | | Weighted median | 0.000611801 | 0.723599391 | 0.601347347 | 0.870704894 |
| | | | Inverse variance weighted | 0.0000131739 | 0.75144733 | 0.660805923 | 0.854521835 |
| | | | Simple mode | 0.293977546 | 0.804099418 | 0.53689448 | 1.204288548 |
| | | | Weighted mode | 0.053483286 | 0.735456442 | 0.541467394 | 0.99894506 |
| ebi-a-GCST90018754 | Total cholesterol levels | 139 | MR Egger | 0.412450582 | 0.832239099 | 0.537165041 | 1.289402445 |
| | | | Weighted median | 0.001077177 | 0.600022601 | 0.441759261 | 0.814984888 |
| | | | Inverse variance weighted | 0.002170221 | 0.73657837 | 0.605805346 | 0.895580896 |
| | | | Simple mode | 0.00502523 | 0.418196564 | 0.229669718 | 0.761477689 |
| | | | Weighted mode | 0.004359167 | 0.553293067 | 0.370814469 | 0.825569777 |
| ieu-a-782 | Total cholesterol | 135 | MR Egger | 0.179557034 | 0.818838391 | 0.612494903 | 1.094696965 |
| | | | Weighted median | 0.202771291 | 0.887220701 | 0.738008358 | 1.066601161 |
| | | | Inverse variance weighted | 0.014166294 | 0.85500402 | 0.754413863 | 0.969006416 |
| | | | Simple mode | 0.671864424 | 1.102694593 | 0.702169846 | 1.731682686 |
| | | | Weighted mode | 0.637025394 | 0.914112915 | 0.630045562 | 1.326257133 |
| ieu-b-4849 | Triglycerides | 89 | MR Egger | 0.005978125 | 0.62604703 | 0.452010731 | 0.867091989 |
| | | | Weighted median | 0.019493486 | 0.828472757 | 0.707471658 | 0.970169054 |
| | | | Inverse variance weighted | 0.001251977 | 0.826188393 | 0.735717911 | 0.927783939 |
| | | | Simple mode | 0.208611554 | 1.318322326 | 0.859619297 | 2.021794719 |
| | | | Weighted mode | 0.631005081 | 1.092184966 | 0.763075994 | 1.563236178 |
| ebi-a-GCST90012082 | N-terminal prohormone brain natriuretic peptide levels | 6 | MR Egger | 0.788167808 | 1.491910137 | 0.097353956 | 22.86292151 |
| | | | Weighted median | 0.007506519 | 1.420002782 | 1.098101689 | 1.836267008 |
| | | | Inverse variance weighted | 0.001624768 | 1.410324814 | 1.138810379 | 1.746573545 |
| | | | Simple mode | 0.08443591 | 1.424625963 | 1.031492598 | 1.967594473 |
| | | | Weighted mode | 0.095355076 | 1.42523893 | 1.016074909 | 1.99916954 |

*(Continued)*

**Table 3.** (Continued)

| id.exposure | Exposure | SNPs | Method | P value | OR | Lower 95%CI | Upper 95%CI |
|---|---|---|---|---|---|---|---|
| ebi-a-GCST005367 | Vitamin D levels | 35 | MR Egger | 0.683150889 | 1.264172287 | 0.414234557 | 3.858035363 |
| | | | Weighted median | 0.043827867 | 1.910543097 | 1.018060885 | 3.585419085 |
| | | | Inverse variance weighted | 0.0000760096 | 2.394771114 | 1.553775729 | 3.690962977 |
| | | | Simple mode | 0.270035856 | 1.9766279 | 0.600670873 | 6.50449028 |
| | | | Weighted mode | 0.191898937 | 1.765414446 | 0.764661666 | 4.075904815 |
| ebi-a-GCST004443 | Interleukin-13 levels | 139 | MR Egger | 0.119615633 | 1.121765321 | 0.971519749 | 1.295246378 |
| | | | Weighted median | 0.002939323 | 1.190584222 | 1.061282521 | 1.335639438 |
| | | | Inverse variance weighted | 0.00000046607 | 1.184044101 | 1.108752649 | 1.264448328 |
| | | | Simple mode | 0.094575574 | 1.248341314 | 0.964196587 | 1.61622231 |
| | | | Weighted mode | 0.008498061 | 1.229915264 | 1.056567366 | 1.431703841 |
| ebi-a-GCST004454 | Interleukin-2 receptor antagonist levels | 107 | MR Egger | 0.010658579 | 0.839400881 | 0.735628793 | 0.957811663 |
| | | | Weighted median | 0.00059236 | 0.819946681 | 0.732138476 | 0.918286064 |
| | | | Inverse variance weighted | 0.001203954 | 0.889222525 | 0.82821925 | 0.954719054 |
| | | | Simple mode | 0.316223361 | 0.882840218 | 0.692706704 | 1.125161408 |
| | | | Weighted mode | 0.009743201 | 0.824355756 | 0.713933087 | 0.951857289 |

SNPs, single nucleotide polymorphisms; OR, Odds ratio; 95%CI, 95% Confidence Interval.

CRP and IL-13, which were associated with other maternal infections, are more strongly linked to systemic inflammatory responses and host defense mechanisms. CRP is a well-established marker of acute inflammation, reflecting the intensity of the innate immune response [42], while IL-13 is involved in Th2-mediated immune pathways and may influence susceptibility to certain infectious processes [43]. Although not retained as primary factors in the MVMR analysis, some protective biomarkers such as HDL and IL-1Ra may still possess potential translational relevance. HDL has been widely recognized for its anti-inflammatory and immunomodulatory properties [44], while IL-1Ra functions as a natural inhibitor of IL-1–mediated inflammatory signaling [45]. These findings suggest that modulation of inflammatory pathways may represent a potential avenue for prevention or intervention. Nevertheless, further studies are required to validate these effects in clinical settings.

Importantly, the MR findings can be partially contextualized within the observed global epidemiological patterns. Regions with higher MSMI burden, particularly low- and middle-SDI countries in South Asia and sub-Saharan Africa, are characterized by increased exposure to infection, poor nutritional status, and limited access to healthcare, all of which may contribute to chronic low-grade inflammation [46–47]. This observation is consistent with our MR findings, which identified inflammatory biomarkers, including CRP, IL-13, and RANTES, as previously reported causal risk factors for MSMI [48–50]. The concordance between epidemiological patterns and genetic evidence suggests that systemic inflammation may represent a key biological pathway linking adverse socioeconomic environments to increased maternal infection risk. Furthermore, in low- and middle-income countries where diagnostic capacity is often limited, the integration of biomarker-based approaches may improve early detection and risk stratification of maternal infections [51]. Thus, these findings suggest that maternal sepsis may be characterized by immune dysregulation and suppression, whereas other maternal infections may be driven more by heightened inflammatory activation. This distinction highlights the potential need for tailored prevention and therapeutic strategies targeting different immune pathways.

The burden estimates presented herein are largely consistent with previous Global Burden of Disease studies, which have shown declining but unevenly distributed maternal infection rates [52]. However, this study adds significant granularity by revealing how socioeconomic stratification affects both absolute and relative burden metrics-a nuance often overlooked in aggregate global analyses. The causal role of inflammatory biomarkers identified through MR extends previous

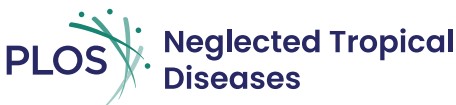

Table 4. MR heterogeneity analyses of associations between variable factors and MSMI.

| id.exposure | Outcome | Exposure | Method | Q-statistic | Q-df | Q P-value |
|---|---|---|---|---|---|---|
| ebi-a-GCST000755 | Maternal sepsis | HDL cholesterol | MR Egger | 132.742861 | 128 | 0.36903889 |
| | | | Inverse variance weighted | 133.274913 | 129 | 0.38030986 |
| ebi-a-GCST005068 | Maternal sepsis | LDL cholesterol | MR Egger | 1.92050296 | 8 | 0.98335275 |
| | | | Inverse variance weighted | 2.43014945 | 9 | 0.98270334 |
| ebi-a-GCST90101747 | Maternal sepsis | Total cholesterol levels | MR Egger | 3.42634417 | 6 | 0.7537421 |
| | | | Inverse variance weighted | 3.47594974 | 7 | 0.83776626 |
| ebi-a-GCST000758 | Maternal sepsis | Triglycerides | MR Egger | 78.9947419 | 141 | 0.99999448 |
| | | | Inverse variance weighted | 80.4237597 | 142 | 0.99999272 |
| ieu-a-783 | Maternal sepsis | Triglycerides | MR Egger | 107.083447 | 163 | 0.99977325 |
| | | | Inverse variance weighted | 107.752613 | 164 | 0.99978168 |
| ieu-b-4850 | Maternal sepsis | Triglycerides | MR Egger | 94.6804383 | 162 | 0.99999455 |
| | | | Inverse variance weighted | 94.9594172 | 163 | 0.99999544 |
| ukb-a-224 | Maternal sepsis | Smoking status: Previous | MR Egger | 132.46237 | 149 | 0.83069857 |
| | | | Inverse variance weighted | 135.00376 | 150 | 0.80445374 |
| ebi-a-GCST004431 | Maternal sepsis | RANTES levels | MR Egger | 93.9900647 | 119 | 0.95609204 |
| | | | Inverse variance weighted | 94.9319698 | 120 | 0.95572199 |
| ebi-a-GCST004443 | Maternal sepsis | Interleukin-13 levels | MR Egger | 127.041642 | 137 | 0.71779248 |
| | | | Inverse variance weighted | 127.101413 | 138 | 0.73686716 |
| ebi-a-GCST004444 | Maternal sepsis | Interleukin-10 levels | MR Egger | 2.94596968 | 5 | 0.70831557 |
| | | | Inverse variance weighted | 2.967531 | 6 | 0.81291087 |
| ebi-a-GCST004447 | Maternal sepsis | Interleukin-1-receptor antagonist levels | MR Egger | 122.007611 | 103 | 0.09741256 |
| | | | Inverse variance weighted | 124.509633 | 104 | 0.08323955 |
| ebi-a-GCST004456 | Maternal sepsis | Interferon gamma levels | MR Egger | 110.356099 | 119 | 0.70228183 |
| | | | Inverse variance weighted | 113.453303 | 120 | 0.65072466 |
| ieu-a-27 | Maternal sepsis | Birth weight | MR Egger | 12.8567604 | 11 | 0.30279053 |
| | | | Inverse variance weighted | 18.2361736 | 12 | 0.10870539 |
| ukb-a-448 | Maternal sepsis | Medication for cholesterol blood pressure diabetes or take exogenous hormones: Cholesterol lowering medication | MR Egger | 3.50251837 | 8 | 0.89899411 |
| | | | Inverse variance weighted | 3.88877634 | 9 | 0.91857919 |
| ukb-b-18009 | Maternal sepsis | Medication for cholesterol, blood pressure, diabetes, or take | MR Egger | 4.40320971 | 18 | 0.99952805 |
| | | | Inverse variance weighted | 4.40747778 | 19 | 0.99977691 |
| prot-a-2400 | Maternal sepsis | Pregnancy-specific beta-1-glycoprotein 11 | MR Egger | 5.61340087 | 18 | 0.9975288 |
| | | | Inverse variance weighted | 7.39602758 | 19 | 0.9917841 |
| prot-a-2408 | Maternal sepsis | Pregnancy-specific beta-1-glycoprotein 9 | MR Egger | 17.3492977 | 31 | 0.97700756 |
| | | | Inverse variance weighted | 17.4167633 | 32 | 0.98310086 |
| ukb-a-355 | Maternal sepsis | Number of pregnancy terminations | MR Egger | 25.8012946 | 26 | 0.47406812 |
| | | | Inverse variance weighted | 25.8630926 | 27 | 0.52620851 |
| ukb-b-12621 | Other maternal infections | Ever had stillbirth, spontaneous miscarriage or termination | MR Egger | 175.286843 | 213 | 0.97225125 |
| | | | Inverse variance weighted | 177.687787 | 214 | 0.96659426 |
| ebi-a-GCST90095034 | Other maternal infections | Body mass index | MR Egger | 26.7993475 | 49 | 0.99590367 |
| | | | Inverse variance weighted | 26.9666765 | 50 | 0.99683189 |
| ieu-a-94 | Other maternal infections | Body mass index | MR Egger | 61.3858909 | 65 | 0.60421154 |
| | | | Inverse variance weighted | 61.9210704 | 66 | 0.61952872 |
| ieu-b-4815 | Other maternal infections | Body mass index | MR Egger | 41.2917887 | 57 | 0.94171176 |
| | | | Inverse variance weighted | 42.4384028 | 58 | 0.93768164 |

*(Continued)*

**Table 4.** (Continued)

| id.exposure | Outcome | Exposure | Method | Q-statistic | Q-df | Q P-value |
|---|---|---|---|---|---|---|
| ebi-a-GCST005067 | Other maternal infections | C-reactive protein levels | MR Egger | 7.22842614 | 15 | 0.95099822 |
| | | | Inverse variance weighted | 7.34060883 | 16 | 0.96612389 |
| ebi-a-GCST90018730 | Other maternal infections | C-reactive protein | MR Egger | 32.8333563 | 51 | 0.97744468 |
| | | | Inverse variance weighted | 33.2407969 | 52 | 0.98009883 |
| ebi-a-GCST000759 | Other maternal infections | LDL cholesterol | MR Egger | 93.3720615 | 164 | 0.99999813 |
| | | | Inverse variance weighted | 95.524593 | 165 | 0.99999679 |
| ebi-a-GCST005068 | Other maternal infections | LDL cholesterol | MR Egger | 1.43418275 | 8 | 0.99374762 |
| | | | Inverse variance weighted | 1.91387229 | 9 | 0.99275746 |
| ebi-a-GCST90018741 | Other maternal infections | LDL cholesterol | MR Egger | 81.2222415 | 83 | 0.53470902 |
| | | | Inverse variance weighted | 83.1995153 | 84 | 0.50416949 |
| ieu-a-781 | Other maternal infections | LDL cholesterol | MR Egger | 60.1375626 | 131 | 0.99999998 |
| | | | Inverse variance weighted | 60.9661753 | 132 | 0.99999998 |
| ieu-b-4845 | Other maternal infections | LDL cholesterol | MR Egger | 32.6061457 | 64 | 0.99962198 |
| | | | Inverse variance weighted | 32.6065236 | 65 | 0.99973666 |
| ieu-a-782 | Other maternal infections | Total cholesterol | MR Egger | 66.6264069 | 133 | 0.99999973 |
| | | | Inverse variance weighted | 66.7309577 | 134 | 0.9999998 |
| ebi-a-GCST90018754 | Other maternal infections | Total cholesterol levels | MR Egger | 128.530834 | 137 | 0.68512589 |
| | | | Inverse variance weighted | 128.904021 | 138 | 0.69834992 |
| ieu-b-4849 | Other maternal infections | Triglycerides | MR Egger | 46.7446912 | 87 | 0.99987418 |
| | | | Inverse variance weighted | 49.9355313 | 88 | 0.99964051 |
| ebi-a-GCST004443 | Other maternal infections | Interleukin-13 levels | MR Egger | 118.312122 | 137 | 0.87370069 |
| | | | Inverse variance weighted | 118.997614 | 138 | 0.87704775 |
| ebi-a-GCST004454 | Other maternal infections | Interleukin-2 receptor antagonist levels | MR Egger | 94.79927 | 105 | 0.75228244 |
| | | | Inverse variance weighted | 95.832303 | 106 | 0.75041729 |
| ebi-a-GCST90012082 | Other maternal infections | N-terminal prohormone brain natriuretic peptide levels | MR Egger | 0.02316797 | 4 | 0.99993342 |
| | | | Inverse variance weighted | 0.02480886 | 5 | 0.99999489 |
| ukb-a-448 | Other maternal infections | Medication for cholesterol blood pressure diabetes or take exogenous hormones: Cholesterol lowering medication | MR Egger | 3.90319485 | 8 | 0.86575018 |
| | | | Inverse variance weighted | 3.90563851 | 9 | 0.91751258 |
| ukb-b-17805 | Other maternal infections | Medication for cholesterol, blood pressure, diabetes, or take exogenous hormones: Cholesterol lowering medication | MR Egger | 18.1140805 | 26 | 0.87158352 |
| | | | Inverse variance weighted | 19.5794325 | 27 | 0.84793373 |
| ukb-b-18009 | Other maternal infections | Medication for cholesterol, blood pressure, diabetes, or take exogenous hormones: Blood pressure medication | MR Egger | 11.2938293 | 18 | 0.88147437 |
| | | | Inverse variance weighted | 11.7175694 | 19 | 0.89734009 |
| ebi-a-GCST005367 | Other maternal infections | Vitamin D levels | MR Egger | 15.8644933 | 33 | 0.99485326 |
| | | | Inverse variance weighted | 17.346877 | 34 | 0.99203558 |

**MSMI, maternal sepsis and maternal other infections; Q,Cochran's Q; Q_df,degrees of freedom for Cochran's Q; Q_pval, p-value of statistical significance for Cochran's Q.**

observational studies that reported associations but could not establish causality [53]. Our finding that HDL cholesterol may be protective offers new insights into potential modifiable factors, complementing experimental evidence suggesting that HDL particles can neutralize bacterial toxins and modulate immune responses. Notably, the divergent projections between ARIMA and BAPC models (Figs 11-12) echo ongoing methodological debates about forecasting infectious

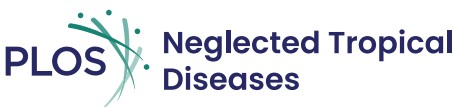

Table 5. MR pleiotropy test of associations between variable factors and MSMI.

| id.exposure | Outcome | Exposure | Egger intercept | Egger P-value | MR-PRESSO RSSobs | MR-PRESSO global P-value |
|---|---|---|---|---|---|---|
| ebi-a-GCST000755 | Maternal sepsis | HDL cholesterol | 0.005672223 | 0.475129483 | 135.5979561 | 0.396 |
| ebi-a-GCST000758 | Maternal sepsis | Triglycerides | 0.009410656 | 0.233931454 | 81.63661094 | 1 |
| ebi-a-GCST005068 | Maternal sepsis | LDL cholesterol | 0.038542093 | 0.495589402 | 2.898157208 | 0.984 |
| ebi-a-GCST90101747 | Maternal sepsis | Total cholesterol levels | -0.008498967 | 0.831139435 | 4.173755419 | 0.852 |
| ieu-a-783 | Maternal sepsis | Triglycerides | 0.006300554 | 0.414536915 | 109.5645232 | 1 |
| ieu-b-4850 | Maternal sepsis | Triglycerides | -0.003925279 | 0.598094071 | 96.24572275 | 1 |
| ukb-a-224 | Maternal sepsis | Smoking status: Previous | -0.025849254 | 0.113016177 | 137.0172496 | 0.786 |
| ebi-a-GCST004431 | Maternal sepsis | RANTES levels | 0.009938388 | 0.333756893 | 96.53206568 | 0.971 |
| ebi-a-GCST004443 | Maternal sepsis | Interleukin-13 levels | 0.001997058 | 0.807223541 | 129.4788713 | 0.734 |
| ebi-a-GCST004444 | Maternal sepsis | Interleukin-10 levels | -0.003905133 | 0.888997096 | 3.894627352 | 0.833 |
| ebi-a-GCST004447 | Maternal sepsis | Interleukin-1-receptor antagonist levels | 0.016208237 | 0.149166153 | 126.8630402 | 0.106 |
| ebi-a-GCST004456 | Maternal sepsis | Interferon gamma levels | -0.014643297 | 0.080996492 | 115.2668184 | 0.681 |
| ieu-a-27 | Maternal sepsis | Birth weight | -0.136888781 | 0.055090675 | 21.09228962 | 0.148 |
| ukb-a-448 | Maternal sepsis | Medication for cholesterol blood pressure diabetes or take exogenous hormones: Cholesterol lowering medication | -0.024727881 | 0.551567389 | 4.801907271 | 0.926 |
| ukb-b-18009 | Maternal sepsis | Medication for cholesterol, blood pressure, diabetes, or take exog-enous hormones: Blood pressure medication | 0.00287889 | 0.948631014 | 4.92575117 | 0.999 |
| prot-a-2400 | Maternal sepsis | Pregnancy-specific beta-1-glycoprotein 11 | 0.041587445 | 0.198468248 | 8.238128917 | 0.993 |
| prot-a-2408 | Maternal sepsis | Pregnancy-specific beta-1-glycoprotein 9 | -0.008816342 | 0.796780532 | 18.53511745 | 0.982 |
| ukb-a-355 | Maternal sepsis | Number of pregnancy terminations | 0.005594817 | 0.805629574 | 27.5872085 | 0.556 |
| ukb-b-12621 | Other maternal infections | Ever had stillbirth, spontaneous mis-carriage or termination | -0.015614573 | 0.122746547 | 179.5733172 | 0.961 |
| ebi-a-GCST90095034 | Other maternal infections | Body mass index | -0.014233343 | 0.684279045 | 28.00369296 | 0.996 |
| ieu-a-94 | Other maternal infections | Body mass index | -0.021032711 | 0.467066726 | 63.53281603 | 0.65 |
| ieu-b-4815 | Other maternal infections | Body mass index | 0.024722583 | 0.288772319 | 43.8810214 | 0.936 |
| ebi-a-GCST005067 | Other maternal infections | C-reactive protein levels | 0.019603322 | 0.742310664 | 8.452549367 | 0.965 |
| ebi-a-GCST90018730 | Other maternal infections | C-reactive protein | 0.011413996 | 0.526127458 | 34.24911822 | 0.985 |
| ebi-a-GCST000759 | Other maternal infections | LDL cholesterol | -0.013729077 | 0.144250098 | 96.71864105 | 1 |
| ebi-a-GCST005068 | Other maternal infections | LDL cholesterol | 0.057373017 | 0.508165114 | 2.204985768 | 0.999 |
| ebi-a-GCST90018741 | Other maternal infections | LDL cholesterol | -0.015208192 | 0.163410733 | 84.91942696 | 0.526 |
| ebi-a-GCST90018754 | Other maternal infections | Total cholesterol levels | -0.006943867 | 0.542284304 | 130.483698 | 0.71 |

*(Continued)*

**Table 5.** (Continued)

| id.exposure | Outcome | Exposure | Egger intercept | Egger P-value | MR-PRESSO RSSobs | MR-PRESSO global P-value |
|---|---|---|---|---|---|---|
| ieu-a-781 | Other maternal infections | LDL cholesterol | 0.008484727 | 0.364345832 | 61.83519629 | 1 |
| ieu-a-782 | Other maternal infections | Total cholesterol | 0.003766704 | 0.746943237 | 67.788827 | 1 |
| ieu-b-4845 | Other maternal infections | LDL cholesterol | -0.000289938 | 0.984549717 | 33.56187092 | 1 |
| ieu-b-4849 | Other maternal infections | Triglycerides | 0.031192006 | 0.077535697 | 51.07842566 | 1 |
| ebi-a-GCST004443 | Other maternal infections | Interleukin-13 levels | 0.010376436 | 0.409141706 | 120.6098671 | 0.894 |
| ebi-a-GCST004454 | Other maternal infections | Interleukin-2 receptor antagonist levels | 0.012829042 | 0.311783622 | 97.39915124 | 0.763 |
| ebi-a-GCST90012082 | Other maternal infections | N-terminal prohormone brain natri-uretic peptide levels | -0.010254851 | 0.969629419 | 0.035985447 | 1 |
| ukb-a-448 | Other maternal infections | Medication for cholesterol blood pressure diabetes or take exogenous hormones: Cholesterol lowering medication | 0.003016256 | 0.96178583 | 4.878109826 | 0.922 |
| ukb-b-17805 | Other maternal infections | Medication for cholesterol, blood pressure, diabetes, or take exoge-nous hormones: Cholesterol lowering medication | 0.036947558 | 0.236970417 | 20.83373081 | 0.866 |
| ukb-b-18009 | Other maternal infections | Medication for cholesterol, blood pressure, diabetes, or take exog-enous hormones: Blood pressure medication | 0.043963533 | 0.523300433 | 12.97198381 | 0.915 |
| ebi-a-GCST005367 | Other maternal infections | Vitamin D levels | 0.026331916 | 0.232036799 | 18.41384171 | 0.988 |

**MR-PRESSO, MR pleiotropy residual sum and outlier; MSMI, maternal sepsis and maternal other infections; RSSobs, residual sum of square observed;MR-PRESSO global P-value, p-value of statistical significance for MR-PRESSO global test.**

diseases. The difference between ARIMA and BAPC results is mainly attributable to their model structures. ARIMA only considers time trends, whereas BAPC incorporates age, period, and cohort effects. Therefore, BAPC may better reflect long-term changes in population structure. Thus, ARIMA is more suitable for short-term prediction, while BAPC provides more reliable long-term estimates. Both models demonstrated good performance in diagnostic tests. The ARIMA model fit the data well based on AIC/BIC and residual tests, while the BAPC model showed stable results in posterior checks. Future studies should prioritize several key areas. First, prospective validation of the identified causal biomarkers in diverse populations is essential, particularly in high-burden regions of Africa and Asia [54]. Research exploring the mechanisms through which lipid metabolism and specific cytokines influence maternal infection susceptibility could reveal new therapeutic targets. Methodologically, developing integrated forecasting models that incorporate climate change, antimicrobial resistance patterns, and healthcare investment scenarios would enhance prediction accuracy. Implementation research examining context-specific strategies for reducing the burden in low-SDI regions is urgently needed-particularly interventions that address both clinical and social determinants of MSMI [55]. Finally, expanding genomic studies to include underrepresented populations will improve the generalizability of causal inferences and help advance precision

**Table 6. MR Steiger directionality test.**

| id.exposure | Outcome | Exposure | snp_r2.expo-sure | snp_r2.outcome | correct_causal_direction | Steiger_pval |
|---|---|---|---|---|---|---|
| ebi-a-GCST000755 | Maternal sepsis | HDL cholesterol | 0.172766974 | 0.00116295 | TRUE | 0 |
| ebi-a-GCST000758 | Maternal sepsis | Triglycerides | 0.260936468 | 0.000789068 | TRUE | 0 |
| ebi-a-GCST005068 | Maternal sepsis | LDL cholesterol | 0.13670689 | 8.77472251754687e-05 | TRUE | 4.48934246922421e-289 |
| ebi-a-GCST90101747 | Maternal sepsis | Total cholesterol levels | 0.032949807 | 7.60439910408359e-05 | TRUE | 4.82117443703251e-138 |
| ieu-a-783 | Maternal sepsis | Triglycerides | 0.375332955 | 0.001008277 | TRUE | 0 |
| ieu-b-4850 | Maternal sepsis | Triglycerides | 0.417524705 | 0.000844268 | TRUE | 0 |
| ukb-a-224 | Maternal sepsis | Smoking status: Previous | 0.018538819 | 0.001157995 | TRUE | 1.14026032633963e-207 |
| ebi-a-GCST004431 | Maternal sepsis | RANTES levels | 0.672725925 | 0.000962158 | TRUE | 0 |
| ebi-a-GCST004443 | Maternal sepsis | Interleukin-13 levels | 0.872902105 | 0.001081431 | TRUE | 0 |
| ebi-a-GCST004444 | Maternal sepsis | Interleukin-10 levels | 0.090442191 | 0.000100461 | TRUE | 1.19379412532228e-143 |
| ebi-a-GCST004447 | Maternal sepsis | Interleukin-1-receptor antago-nist levels | 0.526073036 | 0.001068639 | TRUE | 0 |
| ebi-a-GCST004456 | Maternal sepsis | Interferon gamma levels | 0.294069903 | 0.001054506 | TRUE | 0 |
| ieu-a-27 | Maternal sepsis | Birth weight | 0.018501074 | 0.000224668 | TRUE | 5.71534446473855e-73 |
| ukb-a-448 | Maternal sepsis | Medication for cholesterol blood pressure diabetes or take exogenous hor-mones: Cholesterol lowering medication | 0.010561014 | 9.73078824485904e-05 | TRUE | 2.93924236218571e-139 |
| ukb-b-18009 | Maternal sepsis | Medication for cholesterol, blood pressure, diabetes, or take exogenous hormones: Blood pressure medication | 0.010065199 | 7.14657434896656e-05 | TRUE | 4.24151069540903e-153 |
| prot-a-2400 | Maternal sepsis | Pregnancy-specific beta-1-glycoprotein 11 | 0.13747845 | 0.000169513 | TRUE | 6.93527095697716e-101 |
| prot-a-2408 | Maternal sepsis | Pregnancy-specific beta-1-glycoprotein 9 | 0.285192338 | 0.000303507 | TRUE | 1.50002556130939e-235 |
| ukb-a-355 | Maternal sepsis | Number of pregnancy terminations | 0.011390177 | 0.000256786 | TRUE | 6.17487302983546e-71 |
| ukb-b-12621 | Other maternal infections | Ever had stillbirth, spon-taneous miscarriage or termination | 0.015735548 | 0.00152237 | TRUE | 2.81990950863449e-135 |
| ebi-a-GCST90095034 | Other maternal infections | Body mass index | 0.039895709 | 0.000263659 | TRUE | 1.53542408021117e-290 |
| ieu-a-94 | Other maternal infections | Body mass index | 0.052128823 | 0.000564291 | TRUE | 0 |
| ieu-b-4815 | Other maternal infections | Body mass index | 0.072172921 | 0.00044353 | TRUE | 0 |
| ebi-a-GCST005067 | Other maternal infections | C-reactive protein levels | 0.078534254 | 0.000126633 | TRUE | 3.90595461583497e-149 |
| ebi-a-GCST90018730 | Other maternal infections | C-reactive protein | 0.070119209 | 0.000372273 | TRUE | 0 |
| ebi-a-GCST000759 | Other maternal infections | LDL cholesterol | 0.240555508 | 0.000978787 | TRUE | 0 |
| ebi-a-GCST005068 | Other maternal infections | LDL cholesterol | 0.13670689 | 8.5562584656784e-05 | TRUE | 5.19993733888866e-289 |

*(Continued)*

**Table 6.** (Continued)

| id.exposure | Outcome | Exposure | snp_r2.exposure | snp_r2.outcome | correct_causal_direction | Steiger_pval |
|---|---|---|---|---|---|---|
| ebi-a-GCST90018741 | Other maternal infections | LDL cholesterol | 0.203102753 | 0.000785247 | TRUE | 0 |
| ebi-a-GCST90018754 | Other maternal infections | Total cholesterol levels | 0.134441683 | 0.001151513 | TRUE | 0 |
| ieu-a-781 | Other maternal infections | LDL cholesterol | 0.325527243 | 0.000660266 | TRUE | 0 |
| ieu-a-782 | Other maternal infections | Total cholesterol | 0.250530156 | 0.000605703 | TRUE | 0 |
| ieu-b-4845 | Other maternal infections | LDL cholesterol | 0.317324796 | 0.000429552 | TRUE | 0 |
| ieu-b-4849 | Other maternal infections | Triglycerides | 0.288618339 | 0.000502453 | TRUE | 0 |
| ebi-a-GCST004443 | Other maternal infections | Interleukin-13 levels | 0.872902105 | 0.00120224 | TRUE | 0 |
| ebi-a-GCST004454 | Other maternal infections | Interleukin-2 receptor antagonist levels | 0.722869916 | 0.000885187 | TRUE | 0 |
| ebi-a-GCST90012082 | Other maternal infections | N-terminal prohormone brain natriuretic peptide levels | 0.035021417 | 8.28959349926104e-05 | TRUE | 3.48384787578417e-132 |
| ukb-a-448 | Other maternal infections | Medication for cholesterol blood pressure diabetes or take exogenous hormones: Cholesterol lowering medication | 0.010561014 | 9.02500143124042e-05 | TRUE | 2.02207235991357e-139 |
| ukb-b-17805 | Other maternal infections | Medication for cholesterol, blood pressure, diabetes, or take exogenous hormones: Cholesterol lowering medication | 0.018753757 | 0.000237196 | TRUE | 3.13531143849049e-266 |
| ukb-b-18009 | Other maternal infections | Medication for cholesterol, blood pressure, diabetes, or take exogenous hormones: Blood pressure medication | 0.010065199 | 0.000153414 | TRUE | 1.85145527787838e-139 |
| ebi-a-GCST005367 | Other maternal infections | Vitamin D levels | 0.130750127 | 0.000274771 | TRUE | 0 |

**snp, single nucleotide polymorphisms; Steiger_pval, p value of Steiger directionality test.**

public health approaches to maternal infection prevention. Longitudinal studies tracking how changing SDI modifies MSMI burden could also provide valuable insights for targeting interventions most effectively.

Several limitations warrant consideration when interpreting our results. First, heterogeneity in diagnostic capacity and reporting systems across countries may affect the accuracy of MSMI burden estimates, particularly in low-SDI regions. In low-resource settings, limited laboratory capacity and incomplete surveillance systems may lead to underreporting and misclassification, especially for other maternal infections. Such systematic bias may result in underestimation of the true disease burden and may weaken the observed inverse relationship between SDI and MSMI burden. Despite robust quality control, heterogeneity in diagnostic capability and reporting completeness across countries may affect burden estimates, particularly in low-resource settings [56]. This concern is especially relevant given our finding that diagnostic quality varies with socioeconomic development (Fig 9). Second, the MR analyses assume linearity of causal effects and may not

**Table 7. Results of multivariate mendelian randomization (dimension reduction with LASSO) for maternal sepsis and other maternal infections.**

| Outcome | Exposure | SNPs | P value | OR | Lower 95%CI | Upper 95%CI | Exposure1 | Exposure2 |
|---------|----------|------|---------|-----|-------------|-------------|-----------|-----------|
| Maternal sepsis | RANTES levels | 1 | 0 | 1.44802918 | 1.43913011 | 1.45698328 | RANTES levels | Interleukin-10 levels |
| | Interleukin-10 levels | 2 | 0 | 1.30709749 | 1.30369878 | 1.31050506 | RANTES levels | Interleukin-10 levels |
| Maternal other infections | Interleukin-13 levels | 1 | 0.00656167 | 1.16173727 | 1.04270804 | 1.29435416 | C-reactive protein levels | Interleukin-13 levels |
| | C-reactive protein levels | 4 | 0.00000023385433 | 1.59615066 | 1.33687051 | 1.90571707 | C-reactive protein levels | Interleukin-13 levels |
| Maternal other infections | C-reactive protein levels | 4 | 1.37781487119777e-10 | 1.60932196 | 1.39167948 | 1.86100118 | C-reactive protein levels | NT-proBNP |
| | NT-proBNP | 1 | 0.00000300542091 | 1.53884312 | 1.28421102 | 1.84396342 | C-reactive protein levels | NT-proBNP |

**LASSO, least absolute shrinkage and selection operator; SNPs, single nucleotide polymorphisms; OR, Odds ratio; 95%CI, 95% Confidence Interval; NT-proBNP, N-terminal prohormone brain natriuretic peptide levels.**

capture threshold or non-linear relationships. Although MR reduces confounding, it relies on key assumptions, including the absence of unmeasured confounding of the instrument–outcome association. For complex exposures such as inflammatory cytokines and behavioral traits, genetic instruments may influence outcomes through multiple biological pathways, introducing potential bias [57]. Therefore, despite the application of sensitivity analyses to detect pleiotropy, residual bias from unknown or unmeasured pathways cannot be completely excluded. This limitation has been widely recognized in MR studies involving complex traits. Additionally, the majority of genetic instruments were derived from European-ancestry populations, which may limit the generalizability of the findings to other populations with different genetic architectures [58]. Differences in environmental exposures and gene–environment interactions across populations may further influence the validity of causal estimates. This limitation is particularly relevant for high-burden regions such as Africa and South Asia, where the applicability of the findings may be reduced. The statistical power of the MR analyses may have been insufficient to detect small but clinically meaningful effects, particularly for exposures with a limited number of genome-wide significant SNPs. Although a strict F-statistic threshold (>10) was applied to ensure strong instruments and minimize weak instrument bias, it is important to note that MR studies are inherently constrained by the variance in the exposure explained by the genetic instruments. With a limited number of SNPs, the statistical power to detect modest causal effects is reduced. Thus, null findings should not be interpreted as definitive evidence of no causal relationship, but rather as an indication that any true effect may be smaller than the study was designed to detect. Future investigations leveraging expanded GWAS datasets or employing multi-variable MR approaches to aggregate genetic signals may help overcome these power limitations. It is acknowledged that overlap between exposure and outcome GWAS samples may introduce bias in two-sample MR analyses, particularly by inflating Type I error rates and underestimating standard errors. In this study, this risk was mitigated by selecting exposure and outcome datasets from independent, large-scale consortium efforts. Specifically, the use of UK Biobank-based GWAS for exposures and data from distinct sources (e.g., the FinnGen consortium for outcomes) represents a strategy to minimize overlap, given their differing study populations, ascertainment strategies, and geographic distributions. Therefore, substantial sample overlap is unlikely in this study. Nevertheless, the possibility of partial overlap due to shared control populations or unreported study affiliations cannot be entirely excluded. Future studies utilizing strictly non-overlapping datasets or applying methods to correct for sample overlap (e.g., MRlap, pseudo-IVW) will be valuable to further validate our findings. Third, the forecasting models, though statistically validated,

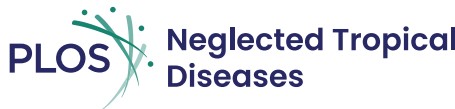

rely on historical trends and may not account for future healthcare breakthroughs, emerging antimicrobial resistance, or climate change impacts on infection patterns [59].

## Conclusions

In summary, this study systematically characterized the global burden and future trends of MSMI and identified key causal risk factors using MVMR. Importantly, by integrating population-level epidemiological analysis with genetic causal inference, this study provides complementary lines of evidence that strengthen the robustness of the findings. The observed discrepancy between temporal trends and absolute burden highlights the critical role of demographic changes, while the identification of inflammatory biomarkers offers potential targets for intervention. To the best of our knowledge, this is the first study to combine GBD-based trend analysis with MVMR in the context of MSMI, thereby providing a novel framework for linking macro-level disease patterns with underlying biological mechanisms. Thus, these findings may inform more precise and evidence-based public health strategies, particularly in high-burden settings.

## Supporting information

**S1 Fig. Spearman's correlation analyses between MSMI Burden and SDI in 1990.** (A) ASIR-SDI, (B) ASPR-SDI, (C) ASMR-SDI, (D) ASDR-SDI, (E) AS-YLD-SDI, and (F) AS-YLL-SDI. DALYs, Disability-Adjusted Life Years; YLDs, Years Lived with Disability; YLLs, Years of Life Lost. ASIR, Age-Standardized Incidence Rate; ASPR, Age-Standardized Prevalence Rate; ASMR, Age-Standardized Mortality Rate; ASDR, Age-Standardized DALYs Rate; AS-YLD, Age-Standardized YLDs Rate; AS-YLL, Age-Standardized YLLs Rate.
(PDF)

**S2 Fig. Spearman's correlation analyses between MSMI Burden and SDI in 2021.** (A) ASIR-SDI, (B) ASPR-SDI, (C) ASMR-SDI, (D) ASDR-SDI, (E) AS-YLD-SDI, and (F) AS-YLL-SDI. DALYs, Disability-Adjusted Life Years; YLDs, Years Lived with Disability; YLLs, Years of Life Lost. ASIR, Age-Standardized Incidence Rate; ASPR, Age-Standardized Prevalence Rate; ASMR, Age-Standardized Mortality Rate; ASDR, Age-Standardized DALYs Rate; AS-YLD, Age-Standardized YLDs Rate; AS-YLL, Age-Standardized YLLs Rate.
(PDF)

**S3 Fig. Decomposition results of MSMI burden across MSMI at the SDI components level.** Three SDI components include population structure (aging), epidemiological changes, and population size (population). DALYs, Disability-Adjusted Life Years; YLDs, Years Lived with Disability; YLLs, Years of Life Lost.
(PDF)

**S4 Fig. Decomposition results of MSMI burden across sex at the SDI components level.** Three SDI components include population structure (aging), epidemiological changes, and population size (population). DALYs, Disability-Adjusted Life Years; YLDs, Years Lived with Disability; YLLs, Years of Life Lost.
(PDF)

**S5 Fig. Leave-one-out plots for the causal association between metabolites and maternal sepsis.** The consistency of results remains robust even after excluding individual genetic variants in each analysis, indicating a high level of reliability and stability in our findings.
(PDF)

**S6 Fig. Leave-one-out plots for the causal association between metabolites and other maternal infections.** The consistency of results remains robust even after excluding individual genetic variants in each analysis, indicating a high level of reliability and stability in our findings.
(PDF)

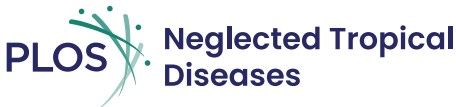

**S1 Table. All original data of MSMI from the GBD database.**
(ZIP)

**S2 Table. All original data of MSMI based on age categories from the GBD database.**
(XLSX)

**S3 Table. All original data of MSMI at the country level from the GBD database.**
(XLSX)

**S4 Table. All original data of MSMI based on age categories and all ages from the GBD database.**
(XLSX)

**S5 Table. All original data of MSMI based on SDI quintiles from the GBD database.**
(XLSX)

**S6 Table. All original data of MSMI based on GBD region categories from the GBD database.**
(XLSX)

**S7 Table. Summary of F-statistics for all exposures in this study.**
(XLSX)

**S8 Table. Summary of coordinated results of instrumental variables (SNPs) for each exposure-outcome pair in this study.**
(XLSX)

**S9 Table. Summary of causal effects using five Mendelian randomization methods in this study.**
(XLSX)

**S10 Table. Summary of horizontal pleiotropy test in Mendelian randomization analysis.**
(XLSX)

**S11 Table. Summary of heterogeneity test in Mendelian randomization analysis.**
(XLSX)

**S12 Table. Summary of Steiger test for causal direction in Mendelian randomization analysis.**
(XLSX)

**S13 Table. Summary of MR-PRESSO global test in Mendelian randomization analysis.**
(XLSX)

**S1 Checklist. The filled checklist is based on the STROBE Statement-Checklist of items that should be included in reports of Mendelian randomization studies (https://www.strobe-mr.org).** *Skrivankova VW, Richmond RC, Woolf BAR, et al. Strengthening the Reporting of Observational Studies in Epidemiology Using Mendelian Randomization: The STROBE-MR Statement. JAMA. 2021;326(16):1614–1621. doi:10.1001/jama.2021.18236.*
(DOCX)

## Author contributions

**Conceptualization:** Anqi Jiang, Siying Duan, Shuyun Wu, Wenkui Yu, Xiaohui Liang.

**Data curation:** Anqi Jiang, Siying Duan, Shuyun Wu.

**Formal analysis:** Anqi Jiang, Siying Duan, Shuyun Wu.

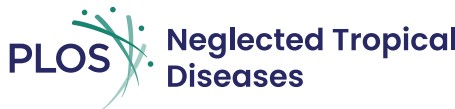

**Investigation:** Anqi Jiang, Siying Duan.

**Methodology:** Anqi Jiang, Siying Duan.

**Project administration:** Anqi Jiang, Siying Duan, Shuyun Wu, Wenkui Yu, Xiaohui Liang.

**Resources:** Siying Duan, Shuyun Wu.

**Supervision:** Wenkui Yu, Xiaohui Liang.

**Visualization:** Anqi Jiang, Siying Duan, Shuyun Wu.

**Writing – original draft:** Anqi Jiang, Xiaohui Liang.

**Writing – review & editing:** Anqi Jiang, Siying Duan, Shuyun Wu, Wenkui Yu, Xiaohui Liang.

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
