## [Decision Letter · Decision Letter 0]

24 Mar 2026

PNTD-D-25-02202

Global Burden, Projections, and Causal factors of Maternal Sepsis and Other Maternal Infections: A Comprehensive Epidemiological and Mendelian Randomization Study

Dear Dr. Liang,

Thank you for submitting your manuscript to PLOS Neglected Tropical Diseases. After careful consideration, we feel that it has merit but does not fully meet PLOS Neglected Tropical Diseases's publication criteria as it currently stands. Therefore, we invite you to submit a revised version of the manuscript that addresses the points raised during the review process.

Please submit your revised manuscript within by May 4 2026 11:59PM. If you will need more time than this to complete your revisions, please reply to this message or contact the journal office at plosntds@plos.org. Please include the following items when submitting your revised manuscript:

We look forward to receiving your revised manuscript.

Kind regards,

Vinícius Silva Belo

Academic Editor

Song Liang

Section Editor

Shaden Kamhawi

co-Editor-in-Chief

Paul Brindley

co-Editor-in-Chief

**Additional Editor Comments:**

Please carefully address all reviewer comments in a comprehensive revision. Particular attention must be given to substantial improvements in the statistical analysis and its reporting (including justification of methods, robustness/sensitivity analyses, model diagnostics, and power considerations), as well as to ensuring that the conclusions are fully supported by the data.

**Journal Requirements:**

At this stage, the following Authors/Authors require contributions: Anqi Jiang, Siying Duan, Shuyun Wu, Wenkui Yu, and Xiaohui Liang. Please ensure that the full contributions of each author are acknowledged in the "Add/Edit/Remove Authors" section of our submission form.

5) Tables should not be uploaded as individual files. Please remove these files and include the Tables in your manuscript file as editable, cell-based objects. For more information about how to format tables, see our guidelines:

https://journals.plos.org/plosntds/s/tables

6) We have noticed that you have uploaded Supporting Information files, but you have not included a list of legends. Please add a full list of legends for your Supporting Information files after the references list.

7) Some material included in your submission may be copyrighted. According to PLOS's copyright policy, authors who use figures or other material (e.g., graphics, clipart, maps) from another author or copyright holder must demonstrate or obtain permission to publish this material under the Creative Commons Attribution 4.0 International (CC BY 4.0) License used by PLOS journals.

Potential Copyright Issues:

- Figure 1: Please confirm whether you drew the images / clip-art within the figure panels by hand. If you did not draw the images, please provide (a) a link to the source of the images or icons and their license / terms of use; or (b) written permission from the copyright holder to publish the images or icons under our CC BY 4.0 license. Alternatively, you may replace the images with open source alternatives. See these open source resources you may use to replace images / clip-art:

- Figures 2, 3, and 10: Please (a) provide a direct link to the base layer of the map (i.e., the country or region border shape) and ensure this is also included in the figure legend; and (b) provide a link to the terms of use / license information for the base layer image or shapefile. We cannot publish proprietary or copyrighted maps (e.g. Google Maps, Mapquest) and the terms of use for your map base layer must be compatible with our CC BY 4.0 license.

**Reviewers' Comments:**

Reviewer's Responses to Questions

**Key Review Criteria Required for Acceptance?**

**Methods**

-Are the objectives of the study clearly articulated with a clear testable hypothesis stated?

-Is the study design appropriate to address the stated objectives?

-Is the population clearly described and appropriate for the hypothesis being tested?

-Is the sample size sufficient to ensure adequate power to address the hypothesis being tested?

-Were correct statistical analysis used to support conclusions?

-Are there concerns about ethical or regulatory requirements being met?

Reviewer #1: -Are the objectives of the study clearly articulated with a clear testable hypothesis stated? NO

-Is the study design appropriate to address the stated objectives? YES

-Is the population clearly described and appropriate for the hypothesis being tested? YES

-Is the sample size sufficient to ensure adequate power to address the hypothesis being tested? NO

-Were correct statistical analysis used to support conclusions? NO

-Are there concerns about ethical or regulatory requirements being met? NO

Reviewer #2: 1. Insufficient consideration of data heterogeneity and robustness validation: Sepsis is a time-dependent condition, and disparities in diagnostic and clinical capacity across healthcare settings and countries may bias global disease burden estimates. While the authors acknowledge this limitation, they did not perform sensitivity analyses or use statistical methods to quantify the impact of such heterogeneity on results, leaving the robustness of conclusions inadequately validated.

2. Limited generalizability of causal inference: Genetic instruments used for Mendelian randomization analyses are predominantly derived from European ancestry populations. Variations in lifestyle, health literacy and cultural contexts across regions can strongly influence sepsis outcomes, yet gene-environment interactions are not explored. These gaps restrict the generalizability of findings to high-burden regions such as Africa and Asia.

3. Insufficient mechanistic discussion and translational relevance: The manuscript lacks in-depth exploration of the molecular mechanisms underlying the associations of lipid metabolism and inflammatory cytokines with maternal infections, and does not integrate existing basic research to support biological plausibility, limiting clinical translational value. Additionally, while targeted interventions are recommended, the study does not propose feasible, evidence-based intervention strategies tailored to key high-risk populations.

4.Academic writing and linguistic issues: The manuscript contains overly lengthy sentences, minor grammatical errors, and imprecise use of academic terminology in several sections, compromising readability and scientific rigor. A thorough linguistic revision is strongly suggested.

**Results**

-Does the analysis presented match the analysis plan?

-Are the results clearly and completely presented?

-Are the figures (Tables, Images) of sufficient quality for clarity?

Reviewer #1: -Does the analysis presented match the analysis plan? YES

-Are the results clearly and completely presented? NO

-Are the figures (Tables, Images) of sufficient quality for clarity? NO

Reviewer #2: (No Response)

**Conclusions**

-Are the conclusions supported by the data presented?

-Are the limitations of analysis clearly described?

-Do the authors discuss how these data can be helpful to advance our understanding of the topic under study?

-Is public health relevance addressed?

Reviewer #1: -Are the conclusions supported by the data presented? NO

-Are the limitations of analysis clearly described? NO

-Do the authors discuss how these data can be helpful to advance our understanding of the topic under study? NO

-Is public health relevance addressed? YES

Reviewer #2: (No Response)

**Editorial and Data Presentation Modifications?**

Reviewer #1: (No Response)

Reviewer #2: (No Response)

**Summary and General Comments**

Reviewer #1: This study addresses a critical topic in maternal and neonatal health by ambitiously integrating Global Burden of Disease (GBD) trend analysis with bidirectional, multivariable Mendelian Randomization (MR). The scope is impressive and the potential public health impact is significant. However, the manuscript in its current form requires substantial revisions to enhance methodological rigor, depth of analysis, logical coherence, and the clarity of its contributions. The epidemiological and MR sections feel disconnected, key methodological details are omitted, and the discussion lacks the mechanistic depth and integrative synthesis needed to fully realize the study's potential.

1. The conflicting projections for ASIR (increasing by ARIMA vs. decreasing by BAPC) represent a core issue. The current discussion is superficial. A thorough analysis explaining this discrepancy is mandatory. Investigate if the ARIMA model failed to capture age-period-cohort effects. I strongly suggest the author to report model fit statistics and model diagnostics. Conduct formal model comparison to justify which projection is more reliable or under which assumptions each trend is plausible.

2. While Figure 6 shows an inverse correlation, formal statistical testing quantifying the SDI-MSMI burden relationship is absent. Furthermore, SDI is a composite index. A decomposition analysis exploring the relative contributions of its components would provide more precise insights into the social determinants.

3. The observation of increasing absolute cases despite declining ASRs in low-SDI regions attributed to population growth needs quantitative support. Please calculate the proportion of the increase attributable to population growth or present overlayed plots of population growth and case numbers by SDI quintile.

4. Figure 10 reveals substantial between-country heterogeneity. This is not addressed in the discussion. Provide case studies or literature references linking these trends to specific national contexts to ground the global analysis.

5. The potential for diagnostic misclassification and under-reporting in low-resource settings is a critical limitation, especially for "other maternal infections." Expand the discussion on how this systemic bias might have underestimated the true burden and potentially attenuated the observed SDI-burden gradient.

6. The F-statistic calculation is mentioned but not reported. This is a key metric for assessing weak instrument bias. Provide a summary of F-statistics for all exposures either in the main text or supplementary materials. The equation for F-statistic calculation should be provided in the Method section.

7. The use of LASSO for variable selection prior to MVMR lacks detail. How was the tuning parameter (λ) selected? Which variables were selected and what were their coefficients? The selection process directly impacts MVMR conclusions and must be transparent.

8. While MR mitigates confounding, the assumption of no genetic confounding is strong. For complex exposures like cytokines and behavioral traits, unmeasured or unknown confounding remains a concern. Strengthen the discussion of this limitation, citing relevant methodological literature.

9. The over-reliance on European genetic data is a major limitation. The discovered causal relationships may not generalize to other populations due to differences in genetic architecture, environment, and gene-environment interactions. Discuss the implications for the applicability of your findings to high-burden regions.

10. There is no mention of whether the MR analyses were sufficiently powered to detect clinically meaningful effect sizes. This is particularly pertinent for exposures with a low number of instrumental SNPs. Consider adding a power.

11. The bias arising from sample overlap between the exposure and outcome populations was not assessed. It is recommended to supplement the analysis with MRlap to correct for the effects of sample overlap.

12. The number of instrumental variables in the Mendelian randomization analysis is too low, with some exposures having as few as 1 SNP and the highest being only 4. This presents a substantial challenge to statistical power. It is difficult to believe that results based on such a limited number of instrumental variables are statistically significant. Therefore, it is advisable to consider appropriately relaxing the criteria for screening instrumental variables and including more SNPs to enhance statistical power.

13. The paper currently reads as two separate stories. A synthesis is lacking. For instance, MR identifies inflammatory markers as key causal factors. Is there epidemiological evidence suggesting higher levels of these markers in high-burden populations/regions?

14. The biological explanations for MR findings are cursory. Why are RANTES and IL-10 independent risks for sepsis, while CRP and IL-13 are risks for other infections? This suggests potentially distinct immunopathological mechanisms. Deepen the discussion by linking these specific cytokines/chemokines to known roles in pregnancy immunology, anti-infective responses, and sepsis pathophysiology.

15. The protective roles of HDL, IL-1Ra, etc., have significant translational potential, which is underexplored. Do these findings point to novel prevention/therapy targets? Could interventions to raise HDL or agents like IL-1Ra analogues be considered for MSMI prevention in high-risk pregnancies? Adopt a more proactive translational medicine perspective.

16. The current abstract and conclusions are largely descriptive. They need to be rewritten to emphasize the value-added of the integrated approach. State clearly that this is the first study to combine GBD trend analysis with MVMR, providing a dual evidence base for future precision public health interventions.

17. Many global maps are cluttered due to the large number of countries, obscuring key messages. Add regional summary insets or tables to global maps. Highlight countries with significantly increasing trends in Fig. 10. Ensure all figure titles and legends are self-explanatory.

18. The acronym "MSMI" is sometimes spelled out. Please use the acronym consistently after its first definition.

19. The results of harmonization and other sensitivity analyses in the Mendelian randomization analysis should be added to the supplementary materials as supplementary results.

20. Descriptions of the features in each column should be added to all tables in the supplementary materials.

21. There are minor grammatical errors and overly long sentences. A thorough language edit is recommended to enhance clarity, conciseness, and precision throughout the manuscript.

Reviewer #2: (No Response)

PLOS authors have the option to publish the peer review history of their article (what does this mean?). If published, this will include your full peer review and any attached files.

Reviewer #1: **Yes:** Yunfeng WANG

Reviewer #2: No

**Figure resubmission:** While revising your submission, we strongly recommend that you use PLOS’s NAAS tool (https://ngplosjournals.pagemajik.ai/artanalysis) to test your figure files. NAAS can convert your figure files to the TIFF file type and meet basic requirements (such as print size, resolution), or provide you with a report on issues that do not meet our requirements and that NAAS cannot fix.
---

## [Decision Letter · Decision Letter 1]

12 May 2026

Dear Dr Liang,

We are pleased to inform you that your manuscript 'Global Burden, Projections, and Causal factors of Maternal Sepsis and Other Maternal Infections: A Comprehensive Epidemiological and Mendelian Randomization Study' has been provisionally accepted for publication in PLOS Neglected Tropical Diseases.

Best regards,

Vinícius Silva Belo

Academic Editor

Song Liang

Section Editor

Shaden Kamhawi

co-Editor-in-Chief

Paul Brindley

co-Editor-in-Chief

The revisions have improved the manuscript, and the final version is suitable for acceptance and publication.

Reviewer's Responses to Questions

**Key Review Criteria Required for Acceptance?**

**Methods**

-Are the objectives of the study clearly articulated with a clear testable hypothesis stated?

-Is the study design appropriate to address the stated objectives?

-Is the population clearly described and appropriate for the hypothesis being tested?

-Is the sample size sufficient to ensure adequate power to address the hypothesis being tested?

-Were correct statistical analysis used to support conclusions?

-Are there concerns about ethical or regulatory requirements being met?

Reviewer #1: -Are the objectives of the study clearly articulated with a clear testable hypothesis stated?

Yes

Is the study design appropriate to address the stated objectives?

Yes

-Is the population clearly described and appropriate for the hypothesis being tested?

Yes

-Is the sample size sufficient to ensure adequate power to address the hypothesis being tested?

Yes

-Were correct statistical analysis used to support conclusions?

Yes

-Are there concerns about ethical or regulatory requirements being met?

Yes

**Results**

-Does the analysis presented match the analysis plan?

-Are the results clearly and completely presented?

-Are the figures (Tables, Images) of sufficient quality for clarity?

Reviewer #1: -Does the analysis presented match the analysis plan?

Yes

-Are the results clearly and completely presented?

Yes

-Are the figures (Tables, Images) of sufficient quality for clarity?

Yes

**Conclusions**

-Are the conclusions supported by the data presented?

-Are the limitations of analysis clearly described?

-Do the authors discuss how these data can be helpful to advance our understanding of the topic under study?

-Is public health relevance addressed?

Reviewer #1: -Are the conclusions supported by the data presented?

Yes

-Are the limitations of analysis clearly described?

Yes

-Do the authors discuss how these data can be helpful to advance our understanding of the topic under study?

Yes

-Is public health relevance addressed?

Yes

**Editorial and Data Presentation Modifications?**

Reviewer #1: NA

**Summary and General Comments**

Reviewer #1: The authors have responded to my concerns. The technical limitations of the article have been described in the Discussion section. Although this study has some seemingly fatal flaws, such as too few instrumental variables and failure to consider sample overlap bias, the authors have provided explanations for these limitations in the Discussion section. Overall, the structure of the article is complete and the statistical methods are rigorous.

PLOS authors have the option to publish the peer review history of their article (what does this mean?). If published, this will include your full peer review and any attached files.

Reviewer #1: **Yes:** Yunfeng WANG

---

## [Editor Report · Acceptance letter]

Dear Postdoc Liang,

We are delighted to inform you that your manuscript, "Global Burden, Projections, and Causal factors of Maternal Sepsis and Other Maternal Infections: A Comprehensive Epidemiological and Mendelian Randomization Study," has been formally accepted for publication in PLOS Neglected Tropical Diseases.

Best regards,

Shaden Kamhawi

co-Editor-in-Chief

Paul Brindley

co-Editor-in-Chief
